# AugGen:
# Synthetic Augmentation using Diffusion Models Can Improve Recognition

**Parsa Rahimi Noshanagh**
EPFL, Idiap
Switzerland
parsa.rahiminoshanagh@epfl.ch

**Damien Teney**
Idiap
Switzerland
damien.teney@idiap.ch

**Sebastien Marcel**
Idiap, UNIL
Switzerland
marcel@idiap.ch

## Abstract

The increasing reliance on large-scale datasets in machine learning poses significant privacy and ethical challenges, particularly in sensitive domains such as face recognition. Synthetic data generation offers a promising alternative; however, most existing methods depend heavily on external datasets or pre-trained models, increasing complexity and resource demands. In this paper, we introduce **AugGen**, a self-contained synthetic augmentation technique. AugGen strategically samples from a class-conditional generative model trained exclusively on the target FR dataset, eliminating the need for external resources. Evaluated across 8 FR benchmarks, including IJB-C and IJB-B, our method achieves **1–12% performance improvements**, outperforming models trained solely on real data and surpassing state-of-the-art synthetic data generation approaches, while using less real data. Notably, these gains often exceed those from architectural enhancements, underscoring the value of synthetic augmentation in data-limited scenarios. Our findings demonstrate that carefully integrated synthetic data can both mitigate privacy constraints and substantially enhance recognition performance. Paper website: https://parsa-ra.github.io/auggen/.

## 1 Introduction

As machine learning increasingly relies on application-specific data, the demand for high-quality, accurately labeled datasets poses significant challenges. Privacy, legal, and ethical concerns amplify these difficulties, particularly in sensitive areas like human face images. A popular solution is synthetic data generation [54, 2, 38], [3], which leverages methods such as 3D-rendering graphics and generative models (*e.g.*, GANs and diffusion models). Notably, synthetic data can surpass real data in model performance, as shown by [54], where 3D-rendered face models with precise labels outperformed real-data-based models in tasks like face landmark localization and segmentation, highlighting the advantages of data synthesis, especially for tasks requiring dense annotations. Image

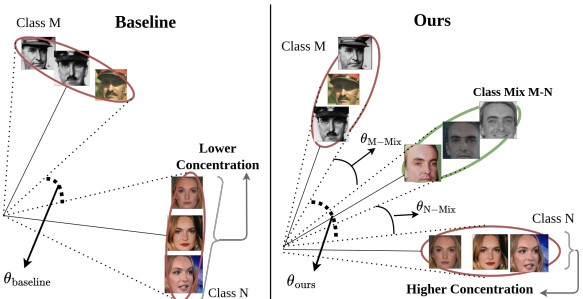

Figure 1: **Core idea of AugGen.** AugGen boosts the model's overall discriminative capabilities without requiring external datasets or pre-trained networks. To achieve this, we propose a novel sampling strategy using a conditional diffusion model—trained exclusively on the discriminator's original data—this enables the generation of synthetic "mixes" of source classes. Incorporating these synthetic samples into the discriminator's training, results in higher intra-class compactness and greater inter-class separation ($\theta_{\text{ours}} > \theta_{\text{baseline}}$) than models trained solely on the original data.

generative models remain underutilized despite rapid advances in VAEs [27], GANs [12, 22, 20], and Diffusion models [47, 19, 21, 17, 13]. Comparisons of generative models often use metrics like Fréchet Distance (FD) [48, 15], which measure similarity to training data, or subjective user preferences for text-to-image tasks [10].

As depicted in Figure 2, currently, synthetic data generation involves training large-scale generative models [39] on datasets such as LAION-5B [43], then refining them via fine-tuning, prompt engineering, or textual inversion [2, 52]. This trend also applies to Face Recognition (FR), where synthetic data aims to mitigate privacy and ethical concerns. However, most methods still rely on large face datasets (which carry their own privacy issues) and auxiliary models, offering no clear advantage over existing real datasets. For instance, DCFace [25] generates diverse face images from multiple identities and uses robust FR systems and auxiliary networks to filter and balance samples. It remains

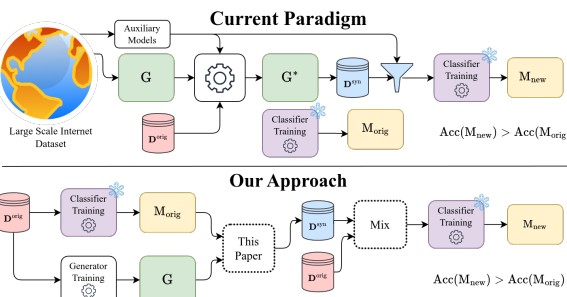

Figure 2: Unlike prior methods that depend on external data or pretrained generators, our self-contained synthetic augmentation framework improves recognition purely through its own generative process.

unclear whether performance gains stem from the datasets, the generative models, or other factors—though larger, more diverse data typically improves results. Contrary to current trends, we advocate using generative models as an augmentation tool for FR training rather than replacing real datasets. Two key factors motivate this stance:

1. Synthetic datasets generated by diffusion models often leak training data [29, 6, 46], offering no clear benefit over existing priors [25, 5, 32, 49, 55].
2. Since responsible FR datasets are scarce and difficult to collect, we aim to boost performance with limited real data, thereby narrowing the gap between small-scale and large-scale training sets.

In this paper, we focus on scenarios involving limited data, demonstrating how we can increase discriminative power by generating synthetic samples while only using a **single labeled dataset**. As illustrated in Figure 1, we generate mixed classes that combine features from two or more source classes while preserving their distinct identities. We choose face recognition (FR) as our primary benchmark due to its unique difficulty, as it requires distinguishing between hundreds of thousands of identities within a highly structured input space. Moreover, FR is a privacy-sensitive task where responsible labeled data are scarce, making it an ideal setting for studying augmentation. Finally, it benefits from a range of well-established benchmarks that enable consistent and meaningful evaluation. To enhance the effect of margin-based losses used by state-of-the-art discriminators in FR systems, given a dataset of $\{(\mathbf{X}, \boldsymbol{y})\}$, where $\mathbf{X}$ is an image and $\boldsymbol{y}$ is its corresponding label, we train a generative model, $p(\mathbf{X} \mid \boldsymbol{y})$, and a discriminator, $p(\boldsymbol{y} \mid \mathbf{X})$, on the same real dataset from scratch. We then introduce a simple yet novel sampling strategy to synthesize new examples. Empirically, we demonstrate that augmenting real data with these carefully generated synthetic samples leads to substantial improvements in the discriminator's performance. Our main contribution is to validate this hypothesis in the context of face recognition (FR):

> **H1**: A generative model can boost the performance of a downstream discriminator with an appropriate informed sampling, and augmenting the resulting data with the original data that was used for training the generative and discriminative models.

Our contributions are summarized as follows:

- We propose a simple yet effective sampling technique that strategically conditions a generative model to produce beneficial samples, enhancing the discriminator's training process (Subsection 3.1) without relying on any auxiliary models/data.
- We show that mixing our AugGen data with real samples **often surpasses even architectural-level improvements**, underscoring that **synthetic dataset generation can be as impactful as architectural advances** (Section 4).

- We demonstrate that AugGen training can be as effective as adding up to **1.7×** real samples, **reducing** the need for more face images while preserving performance (Subsection 4.3).
- We show that current generative metrics (e.g., FD, KD) are poorly correlated with downstream discriminative performance, emphasizing the need for improved proxy metrics (Appendix F).

To the best of our knowledge, this is the first demonstration of generative image models effectively enhancing augmentation at this scale without relying on auxiliary models or external datasets.

## 2 Related Work

**Synthetic Data in Computer Vision.** For a smaller number of class variations, (*e.g.*, 2 or 3 classes for classification target), authors in [11] train separate generative models. This approach is not scalable for a higher number of classes and variations of our target (*e.g.*, we have thousands of classes for training an FR system). In [2], the authors fine-tuned pre-trained diffusion models on ImageNet classes after training on large text-image datasets, demonstrating improved performance on this benchmark through the synthesis of new samples. Authors in [54] leveraged 3D rendering engines and computer graphics. Here as they have access to the underlying 3D Morphable Face Model (3DMM) [4] and closed-form back projection to the image plane, the authors introduced a Face Dataset for landmark detection, localization and also semantic segmentation task. By design, as the method has access to accurate labels in such 3D rendered datasets authors demonstrated a slight advantage on the models trained on their proposed dataset when it is evaluated against real-world datasets.

**Synthetic Data for Face Recognition.** SynFace [37] employs DiscoFaceGAN [9] for controllable identity mixup [57], training with a FR network on MS-Celeb1M [14], 3DMM, keypoint matching, and other priors. DCFace [25] uses dual-condition latent diffusion models (LDMs)—one for style and one for identity—trained on CASIA-WebFace [56], then filters generated images with auxiliary demographic classifiers and a strong FR system. In [45], a StyleGAN2-ADA [23] is pre-trained on a large, unlabeled, multi-ethnic dataset, and an encoder transfers latent-space mappings to an FR network to mitigate bias. GANDiffFace [32] combines StyleGAN3 [20] and Stable Diffusion [39] (trained on LAION-5B [43]), along with DreamBooth [40], for increased intra-class variation. IDiff-face [5] conditions a latent diffusion model on FR embeddings from a network trained on MS1Mv2 [8]. ID$^3$ [55] similarly conditions a diffusion model on face attributes and an FR network trained on MS1Mv2, using both CASIA-WebFace and FFHQ [22] for training. Unlike DCFace's post-processing, ID$^3$ incorporates identity/attribute information directly into the generation process. Note that using MS1Mv2 yields higher FR performance than CASIA-WebFace [8]. DigiFace1M [3] generates diverse 3D-rendered faces with varied poses, expressions, and lighting. In [38], off-the-shelf image-to-image translation [51, 60] further boosts DigiFace1M's performance despite lacking explicit identity information. Additional prior work is discussed in Appendix A.

## 3 Methodology

Figure 3 illustrates our approach, where a discriminator $\mathrm{M}_{\mathrm{orig}}$ and a generator $G$ are trained on the same dataset. By strategically sampling from $G$, we generate synthetic images forming new classes, augmenting the original dataset. We first define the problem for the discriminator and generator in Section 3 and Section 3, then introduce our key contribution: generating new classes (Finding Weights, Figure 3(c)) to complement real datasets with synthetic images.

**Discrimination.** Assume a dataset $\mathbf{D}_{\mathrm{orig}} = \{(\mathbf{X}_i, y_i)\}_{i=0}^{k-1}$, where each $\mathbf{X}_i \in \mathbb{R}^{H \times W \times 3}$ and $y_i \in \{0, \ldots, l-1\}$ ($l < k$). The goal is to learn a discriminative model $f_{\theta_{\mathrm{dis}}} : \mathbf{X} \to \boldsymbol{y}$ that estimates $p(\boldsymbol{y}|\mathbf{X})$ (e.g., on ImageNet [41] or CASIA-WebFace [56]). Typically, similar images have closer features under a measure $m$ (e.g., cosine distance). We train $f_{\theta_{\mathrm{dis}}}$ via empirical risk minimization:

$$\theta_{\mathrm{dis}}^* = \underset{\theta_{\mathrm{dis}} \in \Theta_{\mathrm{dis}}}{\arg\min} \, \mathbb{E}_{(\mathbf{X},y) \sim \mathbf{D}_{\mathrm{orig}}} \big[ \mathcal{L}_{\mathrm{dis}}(f_{\theta_{\mathrm{dis}}}(\mathbf{X}), \boldsymbol{y}) \big], \tag{1}$$

where $\mathcal{L}_{\mathrm{dis}}$ is typically cross-entropy, and $\mathrm{h}_{\mathrm{dis}}$ denotes hyperparameters (e.g., learning rates). The resulting model $\mathrm{M}_{\mathrm{orig}} = f_{\theta_{\mathrm{dis}}^*}$ is shown in Figure 3(a).

**Generative Model.** Generative models seek to learn the data distribution, enabling the generation of new samples. We use diffusion models [47, 1], which progressively add noise to data and train a

denoiser S. Following [19, 21], S is learned in two stages. First, for a given noise level $\sigma$, we add noise $\mathbf{N}$ to $E_{\text{VAE}}(\mathbf{X})$ (or $\mathbf{X}$ directly in pixel-based diffusion) and remove it via:

$$\mathcal{L}(S_{\theta_{den}}; \sigma) = \mathbb{E}_{(\mathbf{X},y)\sim D^{\text{orig}}, \mathbf{N}\sim\mathcal{N}(\mathbf{0},\sigma\mathbf{I})} \left[ \|S_{\theta_{den}}(E_{\text{VAE}}(\mathbf{X}) + \mathbf{N}; c(y), \sigma) - \mathbf{X}\|_2^2 \right], \tag{2}$$

where $c(y)$ denotes the class condition, and $E_{\text{VAE}}(\cdot)$ and $D_{\text{VAE}}(\cdot)$ are optional VAE encoder and decoder. In the second stage, we sample different noise levels and minimize:

$$\theta_{den}^* = \underset{\theta_{den}\in\Theta_{den}}{\arg\min} \ \mathbb{E}_{\sigma\sim\mathcal{N}(\mu,\sigma^2)}\left[\lambda_\sigma \, \mathcal{L}(S_{\theta_{den}}; \sigma)\right], \tag{3}$$

where $\lambda_\sigma$ weights each noise scale. Latent diffusion [39] conducts denoising in a compressed latent space, reducing computational cost for high-resolution data.

## 3.1 Class Mixing

In our formulation, $c$ is one-hot encoded for each label in $D^{\text{orig}}$, then mapped to the denoiser's condition space. After training the conditional denoiser $S_{\theta_{\text{den}}}$ (Figure 3, (c)) via Equation 3, we can sample from the generator in two ways:

1. Use the same one-hot vectors as in training, producing samples similar to $D^{\text{orig}}$. As an example, when passing the one-hot vector for the first class, the generator synthesizes samples that resemble this class (Figure 3, (d)), collectively forming $D^{\text{repro}}$.
2. Apply novel condition vectors $\boldsymbol{c}^*$ different from those used during training.

We explore combining known conditions to synthesize entirely new classes, aiming to increase inter-class separation and feature compactness as presented in Figure 1. By leveraging the previously trained $M_{\text{orig}}$, these additional samples can make $M_{\text{mix}}$ (*i.e.*, discriminator trained on the mix of real and generated data) better across diverse benchmarks.

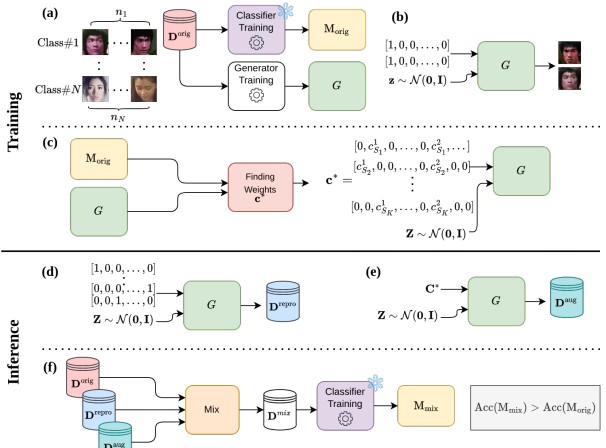

Figure 3: Overview diagram of AugGen: (a) A labeled dataset, $D^{\text{orig}}$, is used to train a class-conditional generator, $G(\mathbf{Z}, \boldsymbol{c})$, and a discriminative model, $M_{\text{orig}}$. (b,d) Reproduced dataset, $D^{\text{repro}}$, closely mimics $D^{\text{orig}}$ under the original conditions. (c) We find new condition vectors, $C^*$, to generate an augmented dataset, $D^{\text{aug}}$, using the generator. (f) Augmenting $D^{\text{orig}}$ with $D^{\text{aug}}$ boosts Morig performance without auxiliary datasets or models.

Given two classes $i$ and $j$ with one-hot vectors $\boldsymbol{c}^{\text{i}}$ and $\boldsymbol{c}^{\text{j}}$, we construct a new class condition via

$$\boldsymbol{c}^* = \alpha\boldsymbol{c}^{\text{i}} + \beta\boldsymbol{c}^{\text{j}}, \tag{4}$$

We denote the trained denoiser's generation process by $G$, so $\mathbf{X}^i = G(\mathbf{Z}, \boldsymbol{c}^i)$ uses noise $\mathbf{Z}\sim\mathcal{N}(\mathbf{0},\mathbf{I})$ and condition $\boldsymbol{c}$ to iteratively denoise the input. To find suitable $\alpha$ and $\beta$, we formulate the problem as a grid search, aiming for dissimilarity to classes $i$ and $j$ while preserving class coherence for repeated samples from $G(\mathbf{Z}, \boldsymbol{c}^*)$. We set the $\alpha$ and $\beta$ to some possible combinations in a linear space of the values between 0.1 to 1.1. Intuitively, the larger either $\alpha$ or $\beta$, the more the generator will reflect the attributes of the corresponding class (*i.e.*, class $i$ and $j$ respectively). For example, possible combinations would be $\alpha = 0.3, \beta = 0.5$ or $\alpha = 1.1, \beta = 0.4$. We denote $\mathbb{W}$, the set which contains possible values of $\alpha$ and $\beta$. We also select some subset of $\mathbb{L}$ and call it $\mathbb{L}_s$, for the set to contain some specific classes. Then we randomly select two values from the $\mathbb{L}_s$, namely $i$ and $j$. Later for each $(\alpha, \beta) \in \mathbb{W}$ we apply the Equation 4, to get the $\boldsymbol{c}^*$. We generate three types of images. The first two is the reproduction dataset, $D^{\text{repro}}$ as before by setting the conditions to $\boldsymbol{c}^i$ and $\boldsymbol{c}^j$, to get $\mathbf{X}^i = G(\mathbf{Z}, \boldsymbol{c}^i)$ and $\mathbf{X}^j = G(\mathbf{Z}, \boldsymbol{c}^j)$. Finally the third one is $\mathbf{X}^* = G(\mathbf{Z}, \boldsymbol{c}^*)$. By passing the generated images to the $f_{\theta_{\text{dis}^*}}$ (*i.e.*, our discriminator which was trained on the $D^{\text{orig}}$) we get the features, $\boldsymbol{e}^i$, $\boldsymbol{e}^j$ and $\boldsymbol{e}^*$ respectively. We seek to maximize the dissimilarity between generated images so that we

can treat the new sample $\mathbf{X}^*$ as a new class. For this, we use a dissimilarity measure, $m_d$ which the *higher* the absolute value it produces the more dissimilar the inputs are. We calculate this measure for each of the reproduced images of the existing classes with respect to the new class, $d_i = m_{\mathrm{d}}(\boldsymbol{e}^i, \boldsymbol{e}^*)$ and $d_j = m_{\mathrm{d}}(\boldsymbol{e}^j, \boldsymbol{e}^*)$, and we define the total dissimilarity between the reproduced classes and the newly generated class as $m_{\mathrm{d}}^{\mathrm{total}} = |d_i| + |d_j|$. We repeat this process $K$ times, this means that we get $K$ different $\mathbf{X}^*$ for the same $\boldsymbol{c}^i$) and $\boldsymbol{c}^j$). We also want each $K$ $\mathbf{X}^*$ to be as similar as possible to each other so we can assign the same label/class to them for a fixed $\alpha$ and $\beta$. To this end, we also calculate a similarity measure, $m_s$, in which the higher the absolute output of this measure is the the more similar their input is. We define the total similarity between the $K$ generated $\mathbf{X}^*$ as $m_{\mathrm{s}}^{\mathrm{total}}$. We hypothesize and verify later with our experiments that the good candidates for $\alpha$ and $\beta$ are the ones that have a high value of the $m^{\mathrm{total}} = m_{\mathrm{s}}^{\mathrm{total}} + m_{\mathrm{d}}^{\mathrm{total}}$. This search for $\alpha$ and $\beta$ is outlined in the algorithm 1.

---

**Algorithm 1:** Grid search for $\alpha$ and $\beta$

**Require:** Search range for $\alpha, \beta \in [0.1, 1.1]$, $\mathbb{L}_s \subseteq \mathbb{L}$, $K$: Number of iterations.
**Require:** $G(.,.)$: Class-conditional Generator trained on $\mathrm{D}^{\mathrm{orig}}$
**Require:** $f_{\theta_{\mathrm{dis}}^*}$: Discriminator trained on $\mathrm{D}^{\mathrm{orig}}$
**Output:** $\alpha^*$ and $\beta^*$

Create set $\mathbb{W} = \{(\alpha, \beta) \mid \alpha, \beta \in [0.1, 1.1]\}$;
Randomly select two values $i$ and $j$ from $\mathbb{L}_s$, $\mathbb{M} = \{\}$ ;
**for** *each* $(\alpha, \beta) \in \mathbb{W}$ **do**
    $\boldsymbol{c}^* = \alpha\boldsymbol{c}^i + \beta\boldsymbol{c}^j$, $\mathbb{M} = \{\}$ ;
    **for** $k = 1, \ldots, K$ **do**
        Get Repro Images: $\mathbf{X}^i = G(\mathbf{Z}, \boldsymbol{c}^i)$, $\mathbf{X}^j = G(\mathbf{Z}, \boldsymbol{c}^j)$;
        Get Interpolated Images: $\mathbf{X}^* = G(\mathbf{Z}, \boldsymbol{c}^*)$;
        Get Repro Features: $\boldsymbol{e}^i, \boldsymbol{e}^j = f_{\theta_{\mathrm{dis}^*}}(\mathbf{X}^i), f_{\theta_{\mathrm{dis}^*}}(\mathbf{X}^j)$;
        Get Interpolated Feature: $\boldsymbol{e}^* = f_{\theta_{\mathrm{dis}^*}}(\mathbf{X}^*)$;
        Add $\boldsymbol{e}^*$ to $\mathbb{F}$;
        Dissimilarities : $d_i = m_{\mathrm{d}}(\boldsymbol{e}^i, \boldsymbol{e}^*)$, $d_j = m_{\mathrm{d}}(\boldsymbol{e}^j, \boldsymbol{e}^*)$;
        Total dissimilarity: $m_{\mathrm{d}}^{\mathrm{total}} = |d_i| + |d_j|$;
    **end**
    $m_{\mathrm{s}}^{\mathrm{total}} = 0$;
    $\forall p, q \in \mathbb{F} | p \neq q$ Calculate $m_{\mathrm{s}}(\boldsymbol{e}^p, \boldsymbol{e}^q)$ and add it to $m_{\mathrm{s}}^{\mathrm{total}}$;
    Final measure: $m^{\mathrm{total}} = m_{\mathrm{s}}^{\mathrm{total}} + m_{\mathrm{d}}^{\mathrm{total}}$ and add it to $\mathbb{M}$;
**end**
Return $\alpha^*$ and $\beta^*$ that the $m^{\mathrm{total}}$, in $\mathbb{M}$ is high;

---

**Algorithm 2:** Generating $\mathrm{D}^{\mathrm{aug}}$

**Require:** $\alpha^*$ and $\beta^*$ from algorithm 1, $\mathbb{L}_s \subseteq \mathbb{L}$, $C$: Number of mixed classes, $N$: Number of samples per class.
**Require:** $G(.,.)$: Class-conditional Generator trained on $\mathrm{D}^{\mathrm{orig}}$
**Output:** $\mathrm{D}^{\mathrm{aug}}$

Create empty set $\mathrm{D}^{\mathrm{aug}}$;
**for** $n = 1, \ldots, C$ **do**
    *Randomly* select two values $i$ and $j$ from $\mathbb{L}_s$;
    $\boldsymbol{c}^* = \alpha^*\boldsymbol{c}^i + \beta^*\boldsymbol{c}^j$;
    Create empty set $T$;
    **for** $n\_samples = 1, \ldots, N$ **do**
        $\mathbf{X}^* = G(\mathbf{Z}, \boldsymbol{c}^*)$;
        Add $\mathbf{X}^*$ to $T$;
    **end**
    Add $T$ to $\mathrm{D}^{\mathrm{aug}}$;
**end**
Return $\mathrm{D}^{\mathrm{aug}}$;

---

After finding candidate values for $\alpha$ and $\beta$, by randomly selecting classes from $\mathbb{L}$, and calculating $\boldsymbol{c}^*$, we can generate images that represent a hypothetically new class. The output of this process is what we call generated augmentations of the $\mathrm{D}^{\mathrm{orig}}$, or $\mathrm{D}^{\mathrm{aug}}$ as depicted in the Figure 3 (e) and presented in algorithm 2. As shown in Figure 1, the newly generated classes are similar within themselves but distinct from their mixed classes, retaining source-class cues to aid discrimination by design. Training with the mix of $\mathrm{D}^{\mathrm{orig}}$ and $\mathrm{D}^{\mathrm{aug}}$ (Figure 3(f)) benefits the discriminator, as demonstrated in Section 4.

## 4 Experiments

We demonstrate the effectiveness of our proposed augmentation method for the problem of Face Recognition (FR). Large datasets are usually required for modern FR systems, so improving performance with limited data is crucial.

### 4.1 Experimental Setup

**Training Data.** We evaluate our approach using two real-world datasets, $\mathrm{D}^{\mathrm{orig}}$: CASIA-WebFace [56] and a subset of WebFace4M [61]. The WebFace4M subset, referred in this work to as WebFace160K, was selected to include approximately 10,000 identities (*i.e.*, like CASIA-WebFace), each

represented by 11 to 24 samples, resulting in a total of  160K face images. More details about the datasets are presented in the Appendix B.

**Discriminative Model.**    To ensure a fair comparison across different methods during the training of the discriminator, we adopted a standardized baseline. This baseline employed an FR system consisting of an IR50 backbone, modified according to the ArcFace's implementation [8], paired with the AdaFace head [24] to incorporate margin loss. Furthermore, when analyzing architectural improvements at the network level, we explored training solely with real data versus mixed data. For this analysis, we used IR101 due to its increased parameterization, which is expected to enhance its ability to generalize. Each real or mixed dataset was trained multiple times with identical hyperparameters but different seed values. More details are outlined in Appendix C. For comparisons, we repeated these procedures using several synthetic datasets from the literature: the original Digi-Face1M (3D graphics), its RealDigiFace translations [38] (Hybrid, 3D, and post-processed), and two diffusion-based datasets, DCFace [25] and IDiff-Face [5]. Additionally, standard augmentations for face recognition tasks were applied to all models. These augmentations included photometric transformations, cropping, and low-resolution adjustments to simulate common variations encountered in real-world scenarios.

**Generative Model.**    To train our generative model, we used a variant of the diffusion formulation [19, 21]. For the $D^{\text{orig}}$ CASIA-WebFace we used the latent-based formulation in which, as depicted in Equation 2 we employed a VAE to encode the image to a compressed space and decode it back to the image space. For WebFace160K we used the pixel space variant for better coverage of different diffusion models. Furthermore, we set the one-hot condition vectors $c^{\sim 10K}$, have a size of $\sim$10,000, corresponding to the number of classes in $D^{\text{orig}}$. We train two versions of the latent diffusion model (LDM) from scratch, labeled small and medium, to analyze the impact of network size and training iterations on the final performance, following the size presets outlined in the original papers [21, 19]. For the pixel-space diffusion model, we mainly used the small variant. Details, including generator design choices are presented in Appendix C.

**Grid Search.**    As presented in the algorithm 1 we need to find an appropriate $\alpha$ and $\beta$ for generating useful augmentations based on the generator trained in the previous section. For the $D^{\text{orig}}$ , CASIA-WebFace which has the long-tail distribution of samples per class, we set the $\mathbb{L}_s$ to the classes from the generator that are presented more than the median number of samples per class. Naturally, we empirically observed that these classes are better reproduced when we were generating $D^{\text{repro}}$. For the case of WebFace160K the $\mathbb{L}_s$ is all the classes. Later we set the $\mathbb{W}$ to $\{0.1, 0.2, \ldots, 1.0, 1.1\}$ for searching $\alpha$ and $\beta$ to calculate the new condition vector $c^*$. Closely related to how the FR models are being trained, especially the usage of the margin loss (*i.e.*, AdaFace [24] or ArcFace [8]), we set the measure for dissimilarity between the features of the two sample images, $\mathbf{X}^1$ and $\mathbf{X}^2$, using cosine similarity to $m_{\text{d}} = 1 - |\frac{e^1 \cdot e^2}{||e^1||||e^2||}|$. Note that the $e$s were calculated using a discriminator that was trained solely on the $D^{\text{orig}}$. We treat the values of the measure in such a way that the higher the output of the measure the more it reflects its functionality (*i.e.*, the larger the measure for dissimilarity is the more dissimilar the inputs are). Accordingly, we set the similarity measure to $m_{\text{s}} = \frac{e^1 \cdot e^2}{||e^1||||e^2||}$, which again reflects that the inputs are more similar if the output of this measure is closer to 1. We iterate multiple choices of the $i$ and $j$ and average our $m^{\text{total}}$ for each of the choices. A sample of the output of this process is depicted in Figure 5. Here we observe that by increasing the $\alpha$ and $\beta$ from $(0.1, 0.1)$ to between $(0.7, 0.7)$ and $(0.8, 0.8)$ the measure increases and after that, it will decrease when we go toward $(1.1, 1.1)$, specifically, we are interested in the $\alpha = \beta$ line as we do not want to include any bias regarding the classes that we **randomly choose**. We consider three sets of values for $(\alpha, \beta)$, $(0.5, 0.5)$, $(0.7, 0.7)$ and $(1.0, 1.0)$ corresponding to the $m^{\text{total}}$ of 1.48, **1.58** and 1.53 respectively. Then the $(\alpha^*, \beta^*)$ respectively from the algorithm 1 for CASIA-WebFace is $(0.7, 0.7)$.

Based on our observations, for the WebFace160K dataset, we performed a coarser parameter search with a higher concentration in the range of 0.5 to 0.9. The total metric value, $m^{\text{total}}$, for WebFace160K is illustrated in the lower part of Figure 5. Using this approach, we evaluated $m^{\text{total}}$ for the parameter pairs $(\alpha, \beta)$ at specific points: $(0.5, 0.5)$, $(0.7, 0.7)$, $(0.8, 0.8)$, and $(1.0, 1.0)$. The corresponding $m^{\text{total}}$ values were 0.6068, 0.7256, **0.7390**, and 0.7230, respectively. Based on these results, the $(\alpha^*, \beta^*)$ pair for WebFace160K was determined to be $(0.8, 0.8)$, as it achieved the highest $m^{\text{total}}$ value of 0.7390. In Appendix G we quantitatively demonstrated the effectiveness of this measure in

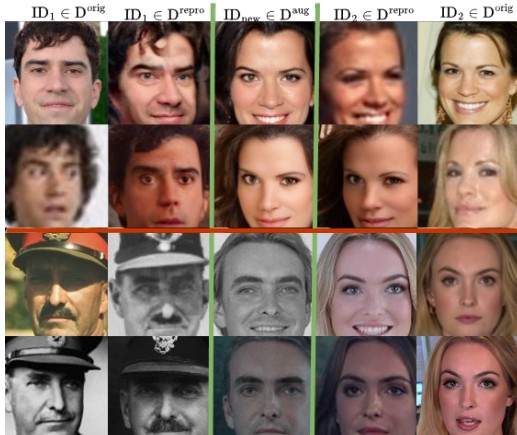

ID₁ ∈ Dᵒʳⁱᵍ  ID₁ ∈ Dʳᵉᵖʳᵒ  IDₙₑw ∈ Dᵃᵘᵍ  ID₂ ∈ Dʳᵉᵖʳᵒ  ID₂ ∈ Dᵒʳⁱᵍ

Figure 4: Randomly sampled images. From left to right: The first column shows variations of a randomly selected identity (ID 1) from $D^{\mathrm{orig}}$. The second column presents the reproduction of the same ID using the generator, conditioned on the corresponding one-hot vector $G(\mathbf{Z}, \boldsymbol{c}_1)$. The third and fourth columns follow the same process for a different ID, with the middle column representing a newly synthesized identity generated by conditioning the generator on $G(\mathbf{Z}, \boldsymbol{c}^*)$. The samples above the red line are from CASIA-WebFace, while the lower part corresponds to WebFace160K.

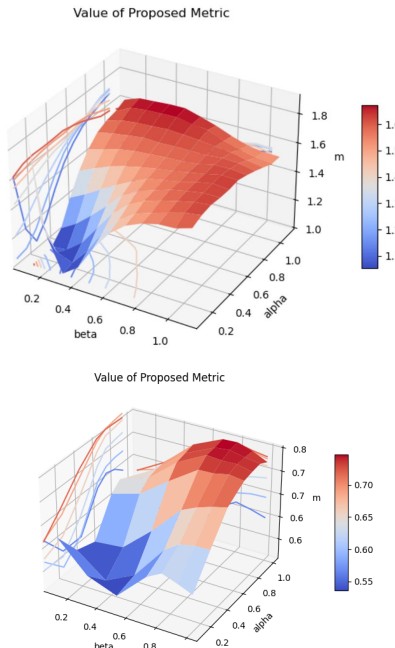

Figure 5: The value of the proposed measure $m^{\mathrm{total}}$ for setting the candidate values of $\alpha$ (x axis) and $\beta$ (y axis). Here for each $\alpha$ and $\beta$ and our 100 combination of $\mathbb{L}_s$ we calculated the $m^{\mathrm{total}}$ by setting the $K$ in algorithm 1 to 10.

the final performance of the discriminator when we trained it on the synthetically generated dataset using various $\alpha$ and $\beta$.

**Computational Complexity.** The search is computationally efficient, requiring fewer than 2 GPU-days on a single consumer-grade GPU (*i.e.*, RTX 3090 Ti in our case), with 1000 mixes (5 samples/class) per grid point. Most compute was spent on repeated training runs for reliable mean/variance reporting. See Subsection C.6 for a detailed compute complexity breakdown.

**Synthetic Dataset.** For generating the reproduction dataset $D^{\mathrm{repro}}$, we set the condition for each of the ~10,000 classes in the original CASIA-WebFace and WebFace160K dataset to the generator. The number of samples per class is 20 unless mentioned otherwise. For generating $D^{\mathrm{aug}}$ we **randomly sampled** $10,000 - 50,000$ combinations of the $\mathbb{L}_s$, $\binom{\mathrm{Card}(\mathbb{L}_s)}{2}$, (samples with more than the median number of sample/class in case of CASIA-WebFace as the $D^{\mathrm{orig}}$), and fixed them for all the experiments. Later by setting the $\alpha$ and $\beta$ to candidate values found in the previous section, (*i.e.*, like $(0.7, 0.7)$ for CASIA), we generated 10 to 50 sample per mixed of selected classes. In Figure 4, some samples of the generated images are shown, where the first and last columns depict examples of the two classes in the $D^{\mathrm{orig}}$. The second and 4-th columns are the reproduction of the same identities from the first and last column, respectively, $D^{repro}$. Each line is generated using the same seed (source of randomness in the generator), and finally, the middle column (3rd from left) is the $D^{\mathrm{aug}}$ which is generated by $\mathbf{X}^* = G(\mathbf{Z}, \boldsymbol{c}^*)$ when we calculate the $\boldsymbol{c}^*$ by optimum $\alpha$ and $\beta$. We can observe that the middle column's identity is slightly different from the source classes while being coherent when we generate multiple examples of this new identity. **By design, these classes can be considered as *hard* examples for the discriminator**. This subtle difference is one of the reasons why our augmentation is improving the final performance. In the Appendix I more samples are presented.

### 4.2 Face Recognition Benchmarks

We show that our synthetic augmentation is boosting the performance of a model trained with the real dataset in all of the studied public FR benchmarks. For this purpose, we evaluated against two sets of FR benchmarks. The first set consists of LFW [18], CFPFP [44], CPLFW [58], CALFW [59], AgeDB

[33], which includes mainly high-quality images with various lighting, poses, and ages the average of these benchmarks presented in Table 1 as **Avg-H**. The second set involves benchmarks consisting of medium to low-quality images from a realistic and more challenging FR scenario (NIST IJB-B/C) [31, 53] and TinyFace [7]. For evaluation, we report verification accuracy (*i.e.*, True Acceptance Rate (TAR)), where the thresholds are set using cross-validation in the high-quality benchmarks, and TARs at different thresholds determined by fixed False Match Rates (FMR) in IJB-B/C. Specifically for the latter, we are mainly interested in the verification accuracy for two thresholds that are usually used in real-world scenarios when the FR systems are being deployed, namely TAR@FPR=1-e-06 and TAR@FPR=1e-05 for both IJB-B and IJB-C. In the Table 1 the **Aux** column depicts that if the method under study used any auxiliary model for the generation of the dataset other than the $D^{orig}$. The ideal value for this column is N which refers to not using any auxiliary model/datasets. The $n^s$ and $n^r$ depict the number of synthetic and real images used for training the discriminative model.

The final values for the benchmarks are reported as the mean and std of the observed numbers when we are changing only the seed as discussed before. Details about the benchmarks, including High-Quality benchmarks and TAR at additional thresholds, are provided in Appendix D. Table 1 is divided into two sections, separated by a triple horizontal line. The upper section compares AugGen, using the CASIA-WebFace dataset as the source, and the lower part is when we set the $D^{orig}$ to WebFace160K. For each, we considered fully synthetic face recognition, $FR_{syn}$, data, and a combination of synthetic and real data, (distinguished by a double horizontal line) $FR_{mix}$. This comparison evaluates their performance relative to the original source dataset (*i.e.,* fully real, $FR_{real}$) and relevant works, including synthetic data from three approaches: the proposed AugGen, DCFace [25], and IDiffFace [5]. The triple horizontal line segmentation is primarily due to the use of CASIA-WebFace and among other data/models in the latter two methods' generation pipelines. For *each* part of the table, **bold** and underline numbers are presenting best and second best respectively. In the second part, in case augmentation with the real CASIA-WebFace is performing better than solely training with the CASIA-WebFace (*i.e.*, middle part of both tables) the cell is shaded in gray . We are observing inconsistencies in different benchmarks for other methods. For instance, for IJB-B/C DCFace is not performing better than CASIA-WebFace alone and IDiffface is not outperforming $FR_{real}$ in thresholds set to low FPRs (*i.e.*, TAR@FPR=1e-6). In the case of $FR_{real}$ training, we additionally used the IR101 network depicted as †. This is done to **demonstrate the introduced augmentation samples can be as important as architectural-level improvements**. As in most cases the less parametrized network (*i.e.*, IR50) trained with the AugGen samples is outperforming the more parametrized network, IR101, solely trained on the original samples, $D^{orig}$. This is in conjunction with the fact that in most cases using the IR101 $FR_{real}$ training outperforms the simpler IR50 model. Additionally, in case our augmentations also perform better than architectural improvements we shade the corresponding cell to green . For the less challenging benchmarks presented by Avg-H in Table 1, we observe that although our method consists of a smaller number of samples and does not use any auxiliary model/data we are performing competitively with other state-of-the-art (SOTA) methods/datasets. In the second part of this table we are observing mainly all the methods that we combined with the CASIA-WebFace are boosting the discriminator which is solely trained on the CASIA-WebFace. For IJB/C we demonstrate better performance being the best in most FPRs although our datasets were generated for augmentation by design. By observing the results after the augmentation (second part of the table), AugGen is the only method that consistently performs better than the baseline. One interesting finding was the performance drop of the model when it was combined with the CASIA-WebFace. But we are observing that *consistently in all of the benchmarks, our augmentation methodology is boosting the baseline*. We demonstrate that although we did not use any auxiliary model/data our synthetic dataset performed competitively with other state-of-the-art methods or even outperformed them in some cases.

The lower part of the triple horizontal line reports results with AugGen samples using our Web-Face160K as the $D^{orig}$. The observations remain the same, as in most cases, we are performing even better than architectural improvements.

As shown in Figure 1, the discriminator's feature space exhibits reduced intra-class variation and increased inter-class separation, with further details in Appendix H.

Table 1: Comparison of the $FR_{syn}$ training (upper part), $FR_{real}$ training (middle), and $FR_{mix}$ training (bottom) using CASIA-WebFace/WebFace160K, when the models are evaluated in terms of accuracy against standard FR benchmarks. **Avg-H** depicts the average accuracy of all high-quality benchmarks including, LFW, CFP-FP, CPLFW, AgeDB, and CALFW. Here $n^s$ and $n^r$ depict the number of Synthetic and Real Images respectively and Aux depicts whether the method for generating the dataset uses an auxiliary information network for generating their datasets (Y) or not (N). the † denotes network trained on IR101 if not the model trained using the IR50. The numbers under columns labeled like C/B-1e-6 indicate TAR for IJB-C/B at FPR of 1e-6. TR1 depicts the rank-1 accuracy for the TinyFace benchmark.

| Method/Data | Aux | $n^s$ | $n^r$ | B-1e-6 | B-1e-5 | C-1e-6 | C-1e-5 | TR1 | Avg-H |
|---|---|---|---|---|---|---|---|---|---|
| DigiFace1M | N/A | 1.22M | 0 | 15.31±0.42 | 29.59±0.82 | 26.06±0.77 | 36.34±0.89 | 32.30±0.21 | 78.97±0.44 |
| RealDigiFace | Y | 1.20M | 0 | 21.37±0.59 | 39.14±0.40 | 36.18±0.19 | 45.55±0.55 | 42.64±1.70 | 81.34±0.02 |
| IDiff-face | Y | 1.2M | 0 | 26.84±2.03 | 50.08±0.48 | 41.75±1.04 | 51.93±0.89 | 45.98±0.61 | 84.68±0.05 |
| DCFace | Y | 1.2M | 0 | 22.48±4.35 | 47.84±6.10 | 35.27±10.78 | 58.22±7.50 | 45.94±0.01 | **91.56±0.09** |
| $D^{aug}$ (Ours) | N | 0.6M | 0 | **29.40±1.36** | **54.54±0.59** | **45.15±1.04** | **61.52±0.47** | 52.33±0.03 | 88.78±0.06 |
| $D^{repro}$ (Ours) | N | 0.6M | 0 | 15.71±3.12 | 45.97±4.64 | 31.54±6.65 | 58.61±3.89 | **53.61±0.47** | 90.64±0.07 |
| CASIA-WebFace | N/A | 0 | 0.5M | 1.02±0.26 | 5.06±1.70 | 0.73±0.19 | 5.37±1.41 | 58.12±0.31 | 94.21±0.09 |
| CASIA-WebFace † | N/A | 0 | 0.5M | 0.74±0.31 | 3.94±1.62 | 0.38±0.13 | 3.92±1.96 | 59.64±0.49 | 94.84±0.07 |
| IDiff-face | Y | 1.2M | 0.5M | 0.89±0.07 | 5.80±0.63 | 0.70±0.11 | 7.46±2.08 | 59.32±0.34 | **94.86±0.02** |
| DCFace | Y | 0.5M | 0.5M | 0.26±0.11 | 1.59±0.51 | 0.18±0.07 | 1.54±0.59 | 56.60±0.41 | 94.72±0.09 |
| $D^{aug}$ (Ours) | N | 0.6M | 0.5M | **2.61±0.91** | **15.74±3.20** | **4.36±1.41** | **18.58±3.99** | **59.82±0.13** | 94.66±0.03 |
| WebFace160K | N/A | 0 | 0.16M | 32.13±1.87 | 72.18±0.18 | 70.37±0.75 | 78.81±0.32 | 61.51±0.16 | 92.50±0.02 |
| WebFace160K † | N/A | 0 | 0.16M | 34.84±0.49 | 74.10±0.24 | 72.56±0.02 | 81.26±0.14 | 62.59±0.01 | 93.32±0.12 |
| $D^{aug}$ (Ours) | N | 0.6M | 0.16M | 36.62±0.77 | 78.32±0.33 | 78.58±0.15 | 85.02±0.15 | 61.60±0.38 | 94.17±0.08 |

Table 2: Effect of adding more real samples from WebFace4M to WebFace160K in comparison to adding more synthetic images. The backbone for all models is IR50. Here **Avg-H** depicts the average accuracy of all high-quality benchmarks including, LFW, CFP-FP, CPLFW, AgeDB, and CALFW. **Ratio** depicts the ratio number of real samples used over the number of samples in WebFace160K. The numbers under columns labeled like C/B-1e-6 indicate TAR for IJB-C/B at FPR of 1e-6.

| Syn #Class × #Sample | $n^r$ | $n^s$ | B-1e-6 | B-1e-5 | C-1e-6 | C-1e-5 | Avg-H | Ratio |
|---|---|---|---|---|---|---|---|---|
| 0 | 160K | 0 | 32.13±1.87 | 72.18±0.18 | 70.37±0.75 | 78.81±0.32 | 92.50±0.02 | 1 |
| (10K x 20 ) | 160K | 200K | 34.93±0.50 | 76.15±0.20 | 75.18±0.22 | 83.06±0.11 | 93.77±0.04 | 1 |
| (20K x 20 ) | 160K | 400K | 36.54±1.27 | 78.00±0.23 | 78.48±0.55 | 84.40±0.07 | 93.96±0.01 | 1 |
| (25K x 20 ) | 160K | 500K | 36.35±0.70 | 77.87±0.52 | **78.61±0.42** | 84.49±0.01 | 94.10±0.08 | 1 |
| (30K x 20 ) | 160K | 600K | **36.62±0.77** | **78.32±0.33** | 78.58±0.15 | 85.02±0.15 | 94.17±0.08 | 1 |
| 0 | 160K + 80K | 0 | 33.78±1.11 | 77.29±0.12 | 77.38±0.10 | 83.50±0.04 | 93.85±0.02 | 1.5 |
| 0 | 160K + 110K | 0 | 33.53±1.47 | 78.26±0.05 | 78.49±0.54 | **85.02±0.01** | **94.19±0.01** | 1.69 |
| 0 | 800K | 0 | 38.12±0.00 | 87.68±0.00 | 87.11±0.00 | 92.27±0.00 | 96.46±0.00 | 5.0 |

## 4.3 Gains over Additional Real Data

In this section, we aim to address a critical question: *How much additional real (non-generated) data would it take to achieve the same performance improvement as our synthetic augmentation?* This experiment is vital because the primary goal is to maximize the accuracy of the face recognition (FR) system using the existing dataset. To evaluate this, we used our *WebFace160K* subset as a baseline and incrementally added data from the WebFace4M dataset. This process allows us to determine how the performance boost achieved through *AugGen* compares to the addition of real data, providing a clear measure of its effectiveness. In Table 2, the **Ratio** represents the proportion of additional real samples added to WebFace160K (*e.g.*, 160K + 110K with a Ratio of 1.69). Remarkably, adding approximately 600K AugGen samples delivers performance gains comparable to including 110K real images. This highlights that AugGen achieves equivalent performance improvements with significantly fewer real images.

## 5 Conclusions

In this work, we introduced *AugGen*, a novel yet simple sampling approach that carefully conditions a generator using a discriminative model, both trained on a single real dataset, to generate augmented samples. By combining these synthetic samples with the original real dataset for training, we enhance the performance of discriminative models without relying on auxiliary data or pre-trained networks. Our proposed AugGen method significantly improves discriminative model performance across 8 FR

benchmarks, consistently outperforming baseline models and, in many cases, exceeding architectural-level enhancements—highlighting its potential to compete with architectural-level improvements. We further demonstrate that training with AugGen-augmented datasets is as effective as using 1.7× more real samples, emphasizing its impact on alleviating data collection challenges. Additionally, we identify inconsistencies in CASIA-WebFace-based evaluations and recommend alternative datasets for more reliable benchmarking on IJB-B/C. Our findings underscore the potential of augmentation-based approaches for improving discriminative models.

**Limitations.** The principal limitation of our approach is its computational cost: to isolate the impact of synthetic data, we train the generator from scratch on the target datasets. Nevertheless, by conducting experiments under these controlled conditions, we establish the hypothesis that synthetic samples generated via our sampling strategy boost the discriminator's performance. Moreover, we expect our method to extend to other architectures (*e.g.*, other multi-step generators, autoregressive, and flows), including pre-trained generators, offering broader practical applicability.

**Future work.** A promising research direction is reformulating margin losses in FR to be compatible with soft labels. By establishing a correlation between target soft labels and $c^*$ (e.g., with $\alpha, \beta = 0.7$ increasing $m^{\text{total}}$, a natural choice for soft target labels would be $0.5, 0.5$ for corresponding source classes), future studies can explore whether treating a class as a soft-class or a new one yields better performance. Also, it would be interesting to see whether the selection process of $\mathbb{L}_s$ will have a major effect on the performance of the models, like mixing some classes will deliver a better performance increase than others.

**Acknowledgment.** This research is based on work conducted in the SAFER project and supported by the Hasler Foundation's Responsible AI program.

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

# Appendix

## A  Summary of SOTA methods

Table 3 summarized recent methodologies for synthetic FR dataset generation. Here the *Generation Methodology* refers to which of the main methods (*i.e.*, Diffusion, GAN, 3DMM, ... ) were used to generate synthetic data. *Auxiliary Networks (Aux)* refers to the use of additional models (e.g., age estimators, face parsers) or datasets during synthetic data generation. The last column, *FR*, indicates whether a strong pre-trained FR backbone, separate from the dataset used for training, was employed or not.

| Method | Year | Generation Methodology | Aux | FR |
|---|---|---|---|---|
| SynFace [37] | 2021 | 3DMM & GAN | Y | Y |
| DigiFace1M [3] | 2023 | 3D-Rendering | Y | N |
| DCFace [25] | 2023 | Diffusion | Y | Y |
| IDiffFace [5] | 2023 | Diffusion | N | Y |
| GANDiffFace [32] | 2023 | GAN/Diffusion | Y | Y |
| RealDigiFace [38] | 2024 | GAN/Diffusion | Y | N |
| ID$^3$[55] | 2024 | Diffusion | Y | Y |
| CemiFace [49] | 2024 | Diffusion | N | Y |
| Ours | - | Diffusion | N | N |

Table 3: State-of-the-art Synthetic Face Recognition (SFR) dataset generation methods are compared based on two criteria: the use of Auxiliary Networks (Aux) and External Face Recognition (FR) Systems. Aux indicates whether auxiliary networks are utilized, with Y representing "Yes" and N representing "No." Similarly, FR highlights the use of external face recognition systems beyond those trained solely on the methodology's dataset, using the same Y/N notation.

## B  Original Datasets $\mathrm{D}^{\mathrm{orig}}$

Table 4 summarizes the key statistics of CASIA-WebFace, WebFace160K, and the original Web-Face4M dataset. Notably, WebFace160K was curated to avoid a long-tail distribution in the number of samples per identity, aligning its statistics more closely to equal presentation while differing from the CASIA-WebFace.

| Name | $n$ IDs | $n^r$ | Min | 25% | 50% | 75% | Max |
|---|---|---|---|---|---|---|---|
| CASIA-WebFace | ∼10.5K | ∼490K | 2 | 18 | 27 | 48 | 802 |
| WebFace160K | ∼10K | ∼160K | 11 | 13 | 16 | 19 | 24 |
| WebFace4M | ∼206K | ∼4,235K | 1 | 6 | 11 | 24 | 1497 |

Table 4: The middle part of the table presents the datasets used in this paper as $\mathrm{D}^{\mathrm{orig}}$, $n$ IDs and $n^r$ representing the number of IDs and real images. The Min and Max present the minimum and maximum number of samples per identity for the corresponding dataset. The number of samples like 25%, 50%, and 75% percentiles is also provided.

## C  Experiment Details

### C.1  Discriminator Training

In the Table 5, the most important parameters for training our discriminative models are presented.

### C.2  Generator Design Choices

Here we try to answer why we are using Diffusion Models and not different types of generators like GANs[15, 23] or VAEs. Theoretically, both VAEs and Diffusion Models train a generator with a maximum likelihood (ML) or ELBO objective; for a detailed derivation, please see [26]. We chose to use a diffusion model primarily because the methodology is more mature, and there are stable empirical procedures for both training (e.g., SNR-based weighting for high-resolution images) and inference

(e.g., faster samplers like DPM-v3). The same can be said for Flow Matching [30]. More specifically, methods like Gaussian Flow Matching (used in Flux and SD3[10]) can be directly formulated as a diffusion model under a v-prediction parameterization. The main difference lies with GANs, whose objective is not formulated as an ELBO or ML. During our experimentation, we attempted to train a StyleGAN-based from scratch on our datasets (CASIA-WebFace and WebFace160K), as no publicly available models were trained on these specific FR datasets, and we aimed to avoid any information leakage from external data like FFHQ. However, as it is well known, GANs are very difficult to train, and our training runs were divergent despite using the settings provided by the original authors. Furthermore, a primary concern with GANs is mode collapse. This makes them an unfavorable choice for our goal, which is to explore out-of-distribution generation. This is especially important for long-tailed datasets like CASIA-WebFace, where modes in the tail would likely not be recovered by a GAN-based generator.

## C.3 Why Grid Search?

## C.4 Generator and Its Training

We trained two sizes of generator, namely small and medium as in [21]. The training of the small-sized generator took about 1 NVIDIA H100 GPU day for the generator to see 805M images in different noise levels with a batch size of $2048$. For reaching the same number of training images for the medium-sized generator, took about 2 days with a batch size of $1024$. We used an Exponential Moving Average (EMA) length of 10%. As observed in literature [35], the EMA of model weights plays a crucial role in the output quality of the Image Generators.

For sampling our models we did **not** employ any Classifier Free Guidance (CFG) [16].

## C.5 Table Details

For the Table 11 we conditioned a medium-sized generator which trained till it saw 805M images in different noise levels ($\sim$1500 Epochs). The conditions were set according to the four sets of values of the $\alpha$ and $\beta$. This is done for a fixed identity combination from the $\mathbb{L}_s$ for all of them. Later for each of these new conditions $c^*$ we generated 50 images. All other tables were reported from a medium-sized generator when they saw 335M training samples.

Table 5: Details of the Discriminator and its Training

| Parameter Name | Discriminator Type 1 | Discriminator Type 2 |
|---|---|---|
| Network type | ResNet 50 | ResNet 50 |
| Marin Loss | AdaFace | AdaFace |
| Batch Size | 192 | 512 |
| GPU Number | 4 | 1 |
| Gradient Acc Step | 1 (For every training step ) | N/A |
| GPU Type | Nvidia RTX 3090 Ti | Nvidia H100 |
| Precision of Floating Point Operations | High | High |
| Matrix Multiplication Precision | High | High |
| Optimizer Type | SGD | SGD |
| Momentum | 0.9 | 0.9 |
| Weight Decay | 0.0005 | 0.0005 |
| Learning Rate | 0.1 | 0.1 |
| WarmUp Epoch | 1 | 1 |
| Number of Epochs | 26 | 26 |
| LR Scheduler | Step | Step |
| LR Milestones | $[12, 24, 26]$ | $[12, 24, 26]$ |
| LR Lambda | 0.1 | 0.1 |
| Input Dimension | $112 \times 112$ | $112 \times 112$ |
| Input Type | RGB images | RGB Images |
| Output Dimension | 512 | 512 |
| Seed | 41,2048,10 (In some models) | 41,2048 |

## C.6 Training time breakdown

The proposed method adds a non-trivial one-time training cost, but this is amortized as it yields a model that is both more accurate and more efficient at inference.

We present a cost breakdown below. Augmenting the data lets the smaller IR-50 backbone outperform the much larger IR-101 model ( 1.9×FLOPs and  1.7×parameters) trained on the original data Table 1.

Crucially, our final model retains the low inference cost of the IR50 backbone while outperforming the IR101 model, which is vital for real-world deployment where cumulative inference costs quickly surpass the one-time training expense.

Table 6: Training times of IR50/IR101 based discriminators on $D^{orig}$ or $D^{orig} + D^{aug}$ datasets next to generator's training time

|  | Train Generator | Train IR50 on $D^{orig}$ | Train IR50 on $D^{orig} + D^{aug}$ (Ours) | Train IR101 on $D^{orig}$ |
|---|---|---|---|---|
| GPU type | 1x H100 | 4x 3090Ti | 4x 3090Ti | 4x 3090Ti |
| Wall time (h) | 42.2 | 2.54 | 4.1 | 5.6 |
| Average perf | N/A | 27.42 ± 0.92 | **32.63 ± 2.20** | 27.24 ± 1.07 |

Variances are calculated as the pooled standard deviation from the results reported in Table 1. This demonstrates a favorable trade-off: we accept a higher, fixed training cost to produce a superior model that is cheaper to deploy.

# D    FR Benchmark Details

The full tables are presented in this section. Detailed results for the High-Quality benchmarks are presented in Table 7. Results for more thresholds set by various FPRs for IJB-B/C are presented in Table 8 and Table 8 respectively.

Table 7: Comparison of the $FR_{syn}$ training (upper part), $FR_{real}$ training (middle), and $FR_{mix}$ training (bottom) using CASIA-WebFace and our WebFace160K, when the models are evaluated in terms of accuracy against standard FR benchmarks, namely LFW, CFPFP, CPLFW, AgeDB and CALFW with their corresponding protocols. Here $n^s$ and $n^r$ depict the number of Synthetic and Real Images respectively and Aux depicts whether the method for generating the dataset uses an auxiliary information network for generating their datasets (Y) or not (N). the † denotes network trained on IR101 if not the model trained using the IR50.

| Method/Data | Aux | $n^s$ | $n^r$ | LFW | CFP-FP | CPLFW | AgeDB | CALFW | Avg |
|---|---|---|---|---|---|---|---|---|---|
| DigiFace1M | N/A | 1.22M | 0 | $92.43_{\pm0.00}$ | $74.64_{\pm0.06}$ | $82.57_{\pm0.43}$ | $75.72_{\pm0.51}$ | $69.48_{\pm1.32}$ | $78.97_{\pm0.44}$ |
| RealDigiFace | Y | 1.20M | 0 | $93.88_{\pm0.19}$ | $76.95_{\pm0.17}$ | $85.47_{\pm0.06}$ | $77.57_{\pm0.07}$ | $72.82_{\pm0.59}$ | $81.34_{\pm0.02}$ |
| IDiff-face | Y | 1.2M | 0 | $97.45_{\pm0.05}$ | $77.07_{\pm0.34}$ | $80.48_{\pm0.63}$ | $87.26_{\pm0.05}$ | $81.15_{\pm0.61}$ | $84.68_{\pm0.05}$ |
| DCFace | Y | 1.2M | 0 | $\mathbf{98.77_{\pm0.12}}$ | $\underline{84.13_{\pm0.35}}$ | $\mathbf{91.19_{\pm0.01}}$ | $\mathbf{92.52_{\pm0.07}}$ | $\mathbf{91.21_{\pm0.06}}$ | $\mathbf{91.56_{\pm0.09}}$ |
| $D^{aug}$ (Ours) | N | 0.6M | 0 | $98.38_{\pm0.12}$ | $83.35_{\pm0.12}$ | $87.64_{\pm0.06}$ | $89.64_{\pm0.29}$ | $84.88_{\pm0.53}$ | $88.78_{\pm0.06}$ |
| $D^{repro}$ (Ours) | N | 0.6M | 0 | $\underline{98.60_{\pm0.02}}$ | $\mathbf{85.26_{\pm0.14}}$ | $\underline{91.13_{\pm0.14}}$ | $\underline{90.54_{\pm0.16}}$ | $\underline{87.69_{\pm0.19}}$ | $\underline{90.64_{\pm0.07}}$ |
| CASIA-WebFace | N/A | 0 | 0.5M | $99.32_{\pm0.02}$ | $88.97_{\pm0.27}$ | $96.35_{\pm0.06}$ | $93.07_{\pm0.13}$ | $93.34_{\pm0.14}$ | $94.21_{\pm0.09}$ |
| CASIA-WebFace † | N/A | 0 | 0.5M | $99.45_{\pm0.05}$ | $89.92_{\pm0.12}$ | $97.06_{\pm0.06}$ | $93.54_{\pm0.02}$ | $94.33_{\pm0.13}$ | $94.86_{\pm0.07}$ |
| IDiff-face | Y | 1.2M | 0.5M | $\mathbf{99.53_{\pm0.07}}$ | $89.92_{\pm0.01}$ | $\mathbf{96.91_{\pm0.27}}$ | $\underline{93.64_{\pm0.16}}$ | $\mathbf{94.28_{\pm0.04}}$ | $\mathbf{94.86_{\pm0.02}}$ |
| DCFace | Y | 0.5M | 0.5M | $99.43_{\pm0.08}$ | $89.44_{\pm0.42}$ | $96.67_{\pm0.16}$ | $\mathbf{93.82_{\pm0.04}}$ | $\underline{94.24_{\pm0.15}}$ | $\underline{94.72_{\pm0.09}}$ |
| $D^{aug}$ (Ours) | N | 0.5M | 0.5M | $\underline{99.47_{\pm0.07}}$ | $\mathbf{89.96_{\pm0.07}}$ | $\underline{96.71_{\pm0.05}}$ | $93.40_{\pm0.22}$ | $93.74_{\pm0.02}$ | $94.66_{\pm0.03}$ |
| WebFace160K | N/A | 0 | 0.16M | $99.08_{\pm0.13}$ | $87.99_{\pm0.45}$ | $93.95_{\pm0.59}$ | $92.75_{\pm0.20}$ | $90.78_{\pm0.79}$ | $92.91_{\pm0.42}$ |
| WebFace160K † | N/A | 0 | 0.16M | $98.97_{\pm0.11}$ | $87.54_{\pm0.06}$ | $93.40_{\pm0.01}$ | $92.55_{\pm0.02}$ | $90.01_{\pm0.04}$ | $92.50_{\pm0.02}$ |
| $D^{aug}$ (Ours) | N | 0.6M | 0.16M | $99.39_{\pm0.03}$ | $89.56_{\pm0.08}$ | $95.84_{\pm0.29}$ | $93.60_{\pm0.10}$ | $92.47_{\pm0.17}$ | $94.17_{\pm0.08}$ |

# E    Mixing Effect

In Table 10, by setting the original dataset to CASIA-WebFace, the effect of increasing the number of samples in our augmented dataset using $(\alpha, \beta) = (0.7, 0.7)$ weights is shown. On average, adding more classes (#Class) and samples per class (#Sample) improves the performance of the final discriminative model. The performance eventually decreases as more samples are added per class. We hypothesize that this is due to the similarity of images generated under the new conditions, $c$, when sampling $G(\mathbf{Z}, c)$ multiple times. This reduces the intra-class variability necessary for training an effective discriminator. We also observe that we should add an appropriate number of the augmentation dataset (*i.e.*, comparing 10k × 5 to without any augmentation) for the final performance to be better than the discriminator trained on the original dataset.

Table 8: Comparison of the $FR_{syn}$ training, $FR_{real}$ training, and $FR_{mix}$ training, when the models are evaluated against IJB-B with thresholds set by various FPRs in terms of TAR. Here $n^s$ and $n^r$ depict the number of Synthetic and Real Images respectively and Aux depicts whether the method for generating the dataset uses an auxiliary information network for generating their datasets (Y) or not (N). the † denotes network trained on IR101 if not the model trained with the IR50. The numbers under columns labeled like *B-1e-6* indicate TAR for IJB-B at FPR of 1e-6.

| Method/Data | Aux | $n^s$ | $n^r$ | B-1e-6 | B-1e-5 | B-1e-4 | B-1e-3 | B-0.01 | B-0.1 | Avg |
|---|---|---|---|---|---|---|---|---|---|---|
| DigiFace1M | N/A | 1.22M | 0 | $15.31_{\pm0.42}$ | $29.59_{\pm0.82}$ | $43.53_{\pm0.77}$ | $59.89_{\pm0.51}$ | $76.62_{\pm0.44}$ | $91.01_{\pm0.12}$ | $52.66_{\pm0.47}$ |
| RealDigiFace | Y | 1.20M | 0 | $21.37_{\pm0.59}$ | $39.14_{\pm0.40}$ | $52.61_{\pm0.70}$ | $67.68_{\pm0.73}$ | $81.30_{\pm0.56}$ | $93.15_{\pm0.17}$ | $59.21_{\pm0.52}$ |
| IDiff-face | Y | 1.2M | 0 | $\underline{26.84}_{\pm2.03}$ | $\underline{50.08}_{\pm0.48}$ | $64.58_{\pm0.32}$ | $77.19_{\pm0.41}$ | $88.27_{\pm0.15}$ | $95.94_{\pm0.05}$ | $67.15_{\pm0.50}$ |
| DCFace | Y | 1.2M | 0 | $22.48_{\pm4.35}$ | $47.84_{\pm6.10}$ | $\mathbf{73.20}_{\pm2.53}$ | $\mathbf{86.11}_{\pm0.59}$ | $93.55_{\pm0.16}$ | $97.56_{\pm0.06}$ | $70.12_{\pm2.28}$ |
| $D^{aug}$ (Ours) | N | 0.6M | 0 | $\mathbf{29.40}_{\pm1.36}$ | $\mathbf{54.54}_{\pm0.59}$ | $70.93_{\pm0.25}$ | $82.95_{\pm0.08}$ | $91.67_{\pm0.10}$ | $97.05_{\pm0.04}$ | $\mathbf{71.09}_{\pm0.11}$ |
| $D^{repro}$ (Ours) | N | 0.6M | 0 | $15.71_{\pm3.12}$ | $45.97_{\pm4.64}$ | $\underline{73.05}_{\pm0.89}$ | $85.54_{\pm0.16}$ | $93.52_{\pm0.17}$ | $\mathbf{97.82}_{\pm0.08}$ | $68.60_{\pm1.43}$ |
| CASIA-WebFace | N/A | 0 | 0.5M | $1.02_{\pm0.26}$ | $5.06_{\pm1.70}$ | $50.37_{\pm4.03}$ | $87.13_{\pm0.38}$ | $95.36_{\pm0.11}$ | $98.36_{\pm0.04}$ | $56.22_{\pm0.99}$ |
| CASIA-WebFace † | N/A | 0 | 0.5M | $0.74_{\pm0.31}$ | $3.94_{\pm1.62}$ | $49.30_{\pm5.75}$ | $88.42_{\pm0.69}$ | $95.78_{\pm0.16}$ | $98.44_{\pm0.09}$ | $56.10_{\pm1.42}$ |
| IDiff-face | Y | 1.2M | 0.5M | $0.89_{\pm0.07}$ | $\underline{5.80}_{\pm0.63}$ | $\underline{54.76}_{\pm2.31}$ | $88.33_{\pm0.49}$ | $\mathbf{96.02}_{\pm0.04}$ | $\mathbf{98.59}_{\pm0.03}$ | $\underline{57.40}_{\pm0.56}$ |
| DCFace | Y | 0.5M | 0.5M | $0.26_{\pm0.11}$ | $1.59_{\pm0.51}$ | $35.62_{\pm7.89}$ | $84.30_{\pm3.52}$ | $95.10_{\pm0.46}$ | $98.36_{\pm0.08}$ | $52.54_{\pm2.08}$ |
| $D^{aug}$ (Ours) | N | 0.5M | 0.5M | $\mathbf{2.61}_{\pm0.91}$ | $\mathbf{15.74}_{\pm3.20}$ | $\mathbf{63.67}_{\pm1.68}$ | $\mathbf{89.19}_{\pm0.28}$ | $\underline{95.78}_{\pm0.02}$ | $98.51_{\pm0.05}$ | $\mathbf{60.92}_{\pm1.02}$ |
| WebFace160K | N/A | 0 | 0.16M | $32.13_{\pm1.87}$ | $72.18_{\pm0.18}$ | $82.96_{\pm0.20}$ | $90.37_{\pm0.04}$ | $95.66_{\pm0.11}$ | $98.75_{\pm0.00}$ | $78.67_{\pm0.40}$ |
| WebFace160K † | N/A | 0 | 0.16M | $34.84_{\pm0.49}$ | $74.10_{\pm0.24}$ | $84.57_{\pm0.41}$ | $91.57_{\pm0.12}$ | $96.09_{\pm0.12}$ | $98.87_{\pm0.03}$ | $80.01_{\pm0.24}$ |
| $D^{aug}$ (Ours) | N | 0.6M | 0.16M | $36.62_{\pm0.77}$ | $78.32_{\pm0.33}$ | $87.65_{\pm0.11}$ | $93.34_{\pm0.13}$ | $96.86_{\pm0.12}$ | $99.01_{\pm0.05}$ | $81.97_{\pm0.16}$ |

Table 9: Comparison of the $FR_{syn}$ training, $FR_{real}$ training, and $FR_{mix}$ training, when the models are evaluated against IJB-C with thresholds set by various FPRs in terms of TAR. Here $n^s$ and $n^r$ depict the number of Synthetic and Real Images respectively and Aux depicts whether the method for generating the dataset uses an auxiliary information network for generating their datasets (Y) or not (N). the † denotes network trained on IR101 if not the model trained with the IR50. The numbers under columns labeled like *B-1e-6* indicate TAR for IJB-C at FPR of 1e-6.

| Method/Data | Aux | $n^s$ | $n^r$ | C-1e-6 | C-1e-5 | C-1e-4 | C-1e-3 | C-0.01 | C-0.1 | Avg |
|---|---|---|---|---|---|---|---|---|---|---|
| DigiFace1M | N/A | 1.22M | 0 | $26.06_{\pm0.77}$ | $36.34_{\pm0.89}$ | $49.98_{\pm0.55}$ | $65.17_{\pm0.39}$ | $80.21_{\pm0.22}$ | $92.44_{\pm0.05}$ | $58.37_{\pm0.46}$ |
| RealDigiFace | Y | 1.20M | 0 | $36.18_{\pm0.19}$ | $45.55_{\pm0.55}$ | $58.63_{\pm0.59}$ | $72.06_{\pm0.90}$ | $84.77_{\pm0.59}$ | $94.57_{\pm0.19}$ | $65.29_{\pm0.50}$ |
| IDiff-face | Y | 1.2M | 0 | $41.75_{\pm1.04}$ | $51.93_{\pm0.89}$ | $65.01_{\pm0.63}$ | $78.25_{\pm0.39}$ | $89.41_{\pm0.19}$ | $96.55_{\pm0.05}$ | $70.48_{\pm0.47}$ |
| DCFace | Y | 1.2M | 0 | $35.27_{\pm10.78}$ | $58.22_{\pm7.50}$ | $77.51_{\pm2.89}$ | $88.86_{\pm0.69}$ | $94.81_{\pm0.09}$ | $98.06_{\pm0.06}$ | $75.46_{\pm3.65}$ |
| $D^{aug}$ (Ours) | N | 0.6M | 0 | $45.15_{\pm1.04}$ | $61.52_{\pm0.47}$ | $74.12_{\pm0.33}$ | $85.09_{\pm0.20}$ | $93.01_{\pm0.17}$ | $97.64_{\pm0.04}$ | $76.09_{\pm0.38}$ |
| $D^{repro}$ (Ours) | N | 0.6M | 0 | $31.54_{\pm6.65}$ | $58.61_{\pm3.89}$ | $78.11_{\pm0.51}$ | $88.51_{\pm0.04}$ | $94.79_{\pm0.05}$ | $98.17_{\pm0.04}$ | $74.96_{\pm1.82}$ |
| CASIA-WebFace | N/A | 0 | 0.5M | $0.73_{\pm0.19}$ | $5.37_{\pm1.41}$ | $56.76_{\pm2.73}$ | $89.44_{\pm0.35}$ | $96.16_{\pm0.07}$ | $98.61_{\pm0.02}$ | $57.84_{\pm0.75}$ |
| CASIA-WebFace † | N/A | 0 | 0.5M | $0.38_{\pm0.13}$ | $3.92_{\pm1.96}$ | $55.21_{\pm6.21}$ | $90.42_{\pm0.76}$ | $96.55_{\pm0.19}$ | $98.69_{\pm0.10}$ | $57.53_{\pm1.54}$ |
| IDiff-face | Y | 1.2M | 0.5M | $0.70_{\pm0.11}$ | $7.46_{\pm2.08}$ | $57.43_{\pm4.17}$ | $89.89_{\pm0.71}$ | $96.63_{\pm0.08}$ | $98.77_{\pm0.01}$ | $58.48_{\pm1.19}$ |
| DCFace | Y | 0.5M | 0.5M | $0.18_{\pm0.07}$ | $1.54_{\pm0.59}$ | $38.17_{\pm8.24}$ | $86.18_{\pm3.32}$ | $95.88_{\pm0.42}$ | $98.59_{\pm0.05}$ | $53.42_{\pm2.11}$ |
| $D^{aug}$ (Ours) | N | 0.5M | 0.5M | $4.36_{\pm1.41}$ | $18.58_{\pm3.99}$ | $67.85_{\pm2.18}$ | $91.12_{\pm0.38}$ | $96.57_{\pm0.07}$ | $98.78_{\pm0.05}$ | $62.88_{\pm1.35}$ |
| WebFace160K | N/A | 0 | ~0.16M | $70.37_{\pm0.75}$ | $78.81_{\pm0.32}$ | $86.45_{\pm0.11}$ | $92.68_{\pm0.01}$ | $96.52_{\pm0.05}$ | $99.02_{\pm0.01}$ | $87.31_{\pm0.20}$ |
| WebFace160K † | N/A | 0 | ~0.16M | $72.56_{\pm0.02}$ | $81.26_{\pm0.14}$ | $88.27_{\pm0.23}$ | $93.55_{\pm0.07}$ | $97.02_{\pm0.07}$ | $99.12_{\pm0.00}$ | $88.63_{\pm0.08}$ |
| $D^{aug}$ (Ours) | N | ~0.6M | ~0.16M | $78.58_{\pm0.15}$ | $85.02_{\pm0.15}$ | $90.87_{\pm0.09}$ | $94.98_{\pm0.09}$ | $97.55_{\pm0.05}$ | $99.23_{\pm0.01}$ | $91.04_{\pm0.04}$ |

Table 10: Effect of mixing different numbers of classes (#Class) and samples per class (#Sample) with the original data, CASIA-WebFace. For TinyFace Rank-1 and Rank-5 verification accuracies are presented as TR1 and TR5 respectively. The numbers under columns labeled like C/B-1e-6 indicate TAR for IJB-C/B at FPR of 1e-6.

| Syn #Class × #Sample | $n^r$ | B-1e-6 | B-1e-5 | C-1e-6 | C-1e-5 | TR1 | TR5 |
|---|---|---|---|---|---|---|---|
| 0 | 0.5M | $1.16_{\pm0.08}$ | $5.61_{\pm1.64}$ | $0.83_{\pm0.10}$ | $5.86_{\pm1.31}$ | $58.01_{\pm0.28}$ | $63.47_{\pm0.07}$ |
| Ours (5k × 5) | 0.5M | $0.85_{\pm0.06}$ | $5.60_{\pm0.84}$ | $0.65_{\pm0.08}$ | $6.70_{\pm0.97}$ | $58.19_{\pm0.20}$ | $63.48_{\pm0.01}$ |
| Ours (5k × 20) | 0.5M | $1.08_{\pm0.16}$ | $5.81_{\pm1.01}$ | $0.84_{\pm0.12}$ | $6.88_{\pm1.38}$ | $57.50_{\pm0.13}$ | $63.07_{\pm0.33}$ |
| Ours (5k × 50) | 0.5M | $0.63_{\pm0.23}$ | $4.56_{\pm0.41}$ | $0.46_{\pm0.10}$ | $6.55_{\pm0.35}$ | $57.39_{\pm0.20}$ | $62.55_{\pm0.11}$ |
| Ours (10K × 5) | 0.5M | $0.77_{\pm0.08}$ | $4.40_{\pm0.14}$ | $0.61_{\pm0.03}$ | $4.69_{\pm0.26}$ | $58.30_{\pm0.28}$ | $63.28_{\pm0.30}$ |
| Ours (10K × 20) | 0.5M | $1.29_{\pm0.01}$ | $8.21_{\pm1.38}$ | $1.43_{\pm0.22}$ | $9.67_{\pm1.01}$ | $58.01_{\pm0.50}$ | $63.00_{\pm0.71}$ |
| Ours (10K × 50) | 0.5M | $0.62_{\pm0.17}$ | $4.29_{\pm0.27}$ | $0.64_{\pm0.10}$ | $5.98_{\pm0.00}$ | $57.51_{\pm0.32}$ | $62.77_{\pm0.08}$ |

# F  Downstream Performance vs Metrics in Generative Models

In this section, we examine whether there is a correlation between common metrics for evaluating generative models and the discriminator's performance when trained on our augmented dataset. We studied the FD [15] Precision/Recall [42, 28] and Coverage [34] which is usually used to quantify the performance of the Generative Models. Calculation of these metrics requires the projection

of the images into meaningful feature spaces. For feature extraction, we consider two backbones, Inception-V3 [50] and DINOv2 [36] which the latter shown effective for evaluating diffusion models [48]. Both these models were trained using the ImageNet [41] in a supervised and semi-supervised manner respectively. Experiments were performed by randomly selecting $100,000$ images of both CASIA-WebFace (as the source distribution) and our generated images by the value of $\alpha$ and $\beta$ using algorithm 1 (*i.e.*, the same settings as presented in the Section 4). We are reporting four versions of our generated augmentation using a medium-sized generator when it sees 184M, 335M, 603M, and 805M training samples in different noise scales of the original CASIA-WebFace (M for Million). For each of the classes generated from these models, we selected 20 samples, based on the previous observation in Table 10. Later by mixing the selected images with the original CASIA-WebFace we train FR for each of them and report the average accuracies for different thresholds in the IJB-C (*i.e.*, similar to **Avg** column in the Table 9). Figure 6 and Figure 7 are showing mentioned metrics for Inception-V3 and DINOv2 feature extractor respectively. We observe no clear correlation between the metrics used to evaluate generative models and the performance of a downstream task. When comparing $D^{aug}$ to $D^{orig}$ for FD, a higher FD (*i.e., distinguishable* $D^{aug}$ images) should enhance discriminator performance, but that wasn't observed here. This holds when we are augmenting the dataset for training the generator and discriminator with the $D^{orig}$. This highlights the need to develop new evaluation metrics as a proxy.

# G  Effectiveness of Grid Search

We also study the effectiveness of our proposed method in algorithm 1 which tries to find the suitable condition weights, $\alpha$, and $\beta$. We compare with four sets of values:

- Rand: $\alpha$ and $\beta$ were selected randomly for $10,000$ mixture of identities from the set of $\{0.1, 0.3, 0.5, 0.7, 0.9, 1.0, 1.1\}$.
- Half: $\alpha$ and $\beta$ set to 0.5 for all $10,000$ random mixture of identities selected from $\mathbb{L}_s$.
- Full: $\alpha$ and $\beta$ set to 1 for all $10,000$ random mixture of identities selected from $\mathbb{L}_s$.
- Half++: $\alpha$ and $\beta$ set to 0.7 according to the algorithm 1 for the generator and discriminator trained on CASIA-WebFace dataset. This is done for all $10,000$ random mixture of identities selected from $\mathbb{L}_s$

The results for this are shown in the Table 11. We observe on almost all of the benchmarks the $D^{aug}$ generated using $\alpha$ and $\beta$ values with higher $m^{\text{total}}$ are performing better.

Table 11: Effectiveness of our weighting procedure (W/ Half++) in comparison to (W/ Random) or when putting the conditions to 0.5 (W/ Half) and when setting the condition signal to 1 (W/ Full). Best in bold, second best, underlined. TR1 represents the Rank-1 accuracy for the TinyFace benchmark. The numbers under columns labeled like C/B-1e-6 indicate TAR for IJB-C/B at FPR of 1e-6

| C Weight Method | $n^s$ | $n^r$ | B-1e-6 | B-1e-5 | C-1e-6 | C-1e-5 | TR1 | $m^{\text{total}}$ |
|---|---|---|---|---|---|---|---|---|
| W/ Half | $\sim$0.5M | 0 | $8.52_{\pm5.61}$ | $27.74_{\pm6.87}$ | $11.59_{\pm4.26}$ | $35.69_{\pm5.23}$ | $46.42_{\pm0.60}$ | 1.48 |
| W/ Full | $\sim$0.5M | 0 | $17.63_{\pm0.08}$ | $32.47_{\pm0.47}$ | $24.30_{\pm0.80}$ | $37.45_{\pm0.22}$ | $45.08_{\pm0.17}$ | 1.53 |
| W/ Random | $\sim$0.5M | 0 | $\underline{24.47_{\pm1.23}}$ | $\underline{39.83_{\pm1.08}}$ | $\underline{30.79_{\pm1.39}}$ | $\underline{44.33_{\pm0.88}}$ | $\mathbf{49.34_{\pm0.31}}$ | N/A |
| W/ Half++ | $\sim$0.5M | 0 | $\mathbf{25.44_{\pm0.19}}$ | $\mathbf{46.20_{\pm0.12}}$ | $\mathbf{39.66_{\pm0.38}}$ | $\mathbf{51.47_{\pm0.29}}$ | $\underline{47.95_{\pm0.09}}$ | $\mathbf{1.58}$ |

# H  Verifying the driving Hypothesis

As shown in Figure 1, introducing a new class using algorithm 2, aims to augment the original dataset with a novel mix of source classes. This approach enforces the network to improve the compactness and separability of class representations. By requiring the network to distinguish the mixed class from its source classes, we strengthen its discriminative power. To validate this approach, we conducted experiments on two models, $f_{\theta_{\text{dis}}}^{\text{Baseline}}$ and $f_{\theta_{\text{dis}}}^{\text{AugGen}}$, trained before and after incorporating AugGen samples, respectively, and evaluated their performance using the following metrics:

1. **Mean** absolute Inter-Class Similarity of samples across all mixed classes. After applying AugGen, we expect that the average similarity of samples from different classes become lower, corresponding to a higher $\theta_{\text{ours}}$ in Figure 1.

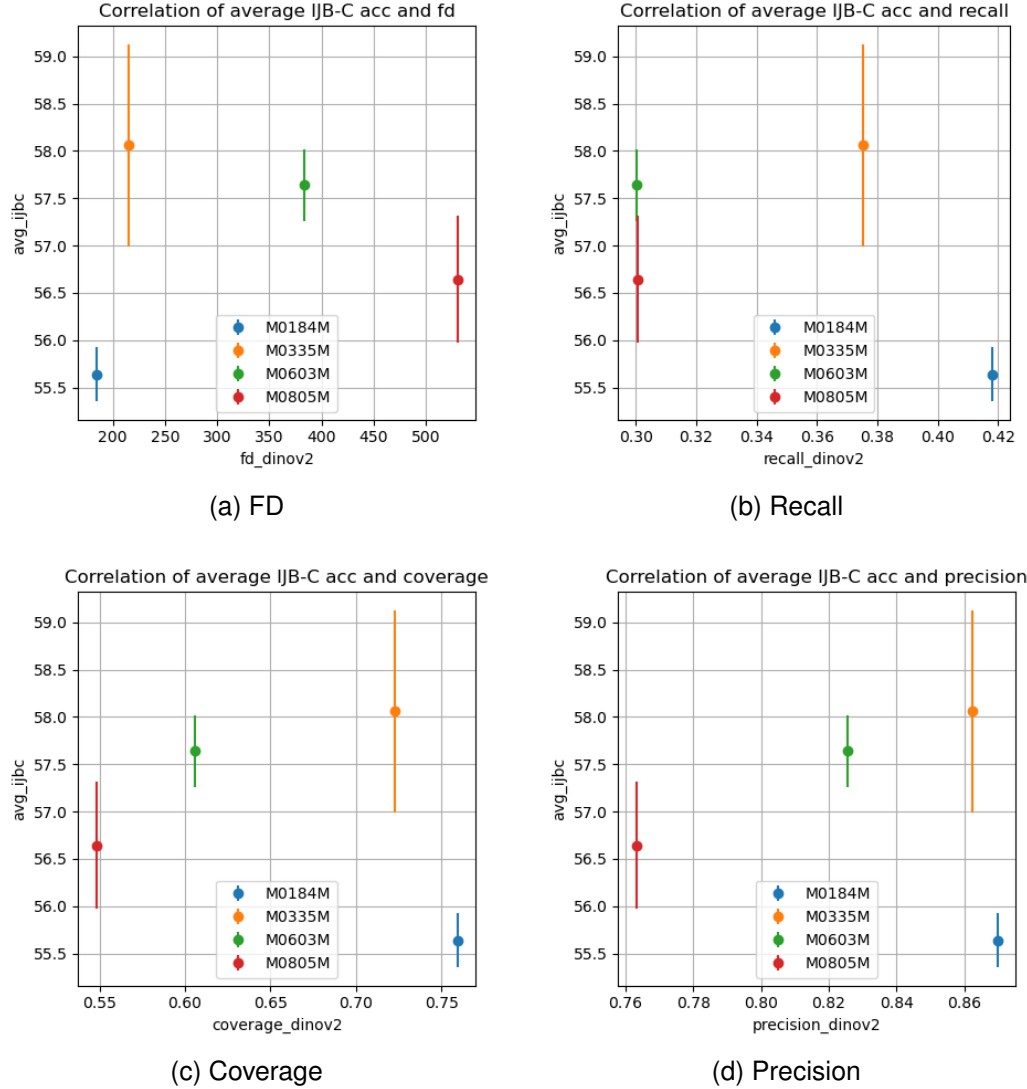

Figure 6: Correlation between the FD, Recall, Coverage, and Precision for the generated dataset by comparing it with the features of CASIA-WebFace using the DINOv2 extractor.

2. **Mean** and standard deviation (*i.e.*, **std**) of Intra-Class Similarity of samples of all mixed classes, (*i.e.*, M-Intra and S-Intra in Table 12). This should indicate if the generated samples for each class, cause the model to boost its compactness.

These metrics are presented in the Table 12. After adding the AugGen samples, we are observing lower M-Inter which reflects that the similarity of the samples between different classes decreased. We are also observing the M-Intra increase reflecting that the networks perceive the images of the same class as more similar.

Table 12: Comparison of models trained with and without *AugGen* samples: *M-Inter* represents interclass similarity, indicating class separation, while *M-Intra* and *S-Intra* measure the mean and standard deviation of intraclass similarity, reflecting class compactness.

| Dataset/Method | $n^s$ | $n^r$ | M-Inter($\downarrow$) | M-Intra($\uparrow$) | S-Intra($\downarrow$) |
|---|---|---|---|---|---|
| Baseline | 0 | 0.16M | 0.0672 | 0.49065 | 0.13499 |
| AugGen | 0.2M | 0.16M | **0.0664** | **0.54917** | **0.12807** |

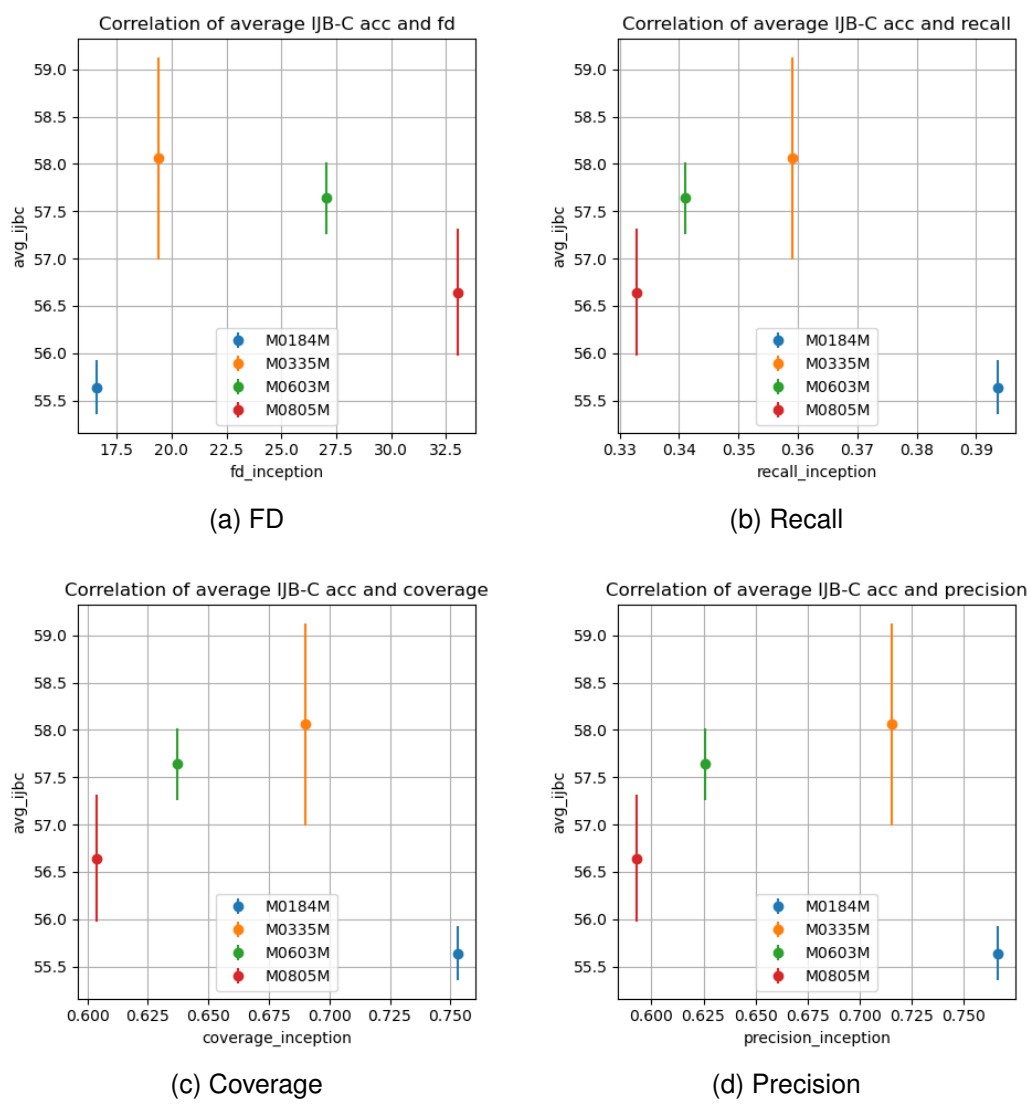

Figure 7: Correlation between the FD, Recall, Coverage, and Precision for the generated dataset by comparing it with the features of CASIA-WebFace using Inception-v3 extractor.

# I   More Samples of $D^{aug}$

In the following figures, you can find more examples of generated images for Small and Medium-sized generators and also trained for more steps. By comparing Figure 8 (generated result from a small-sized generator trained when it sees 335M images ($\sim$ 700 Epochs), **S335M**, as the optimization of score-function, involves multiple noise levels of images), Figure 9 (**M335M**) and Figure 13 (**M805M**) we generally observe that larger generators are producing better images, but training for more steps does not necessarily translate to better image quality. This is especially important as we are exploring the out-of-distribution generation capabilities of an image generator.

## Reproducibility.

All code for the discriminative and generative models, along with the generated datasets and trained models, will be publicly available for reproducibility.

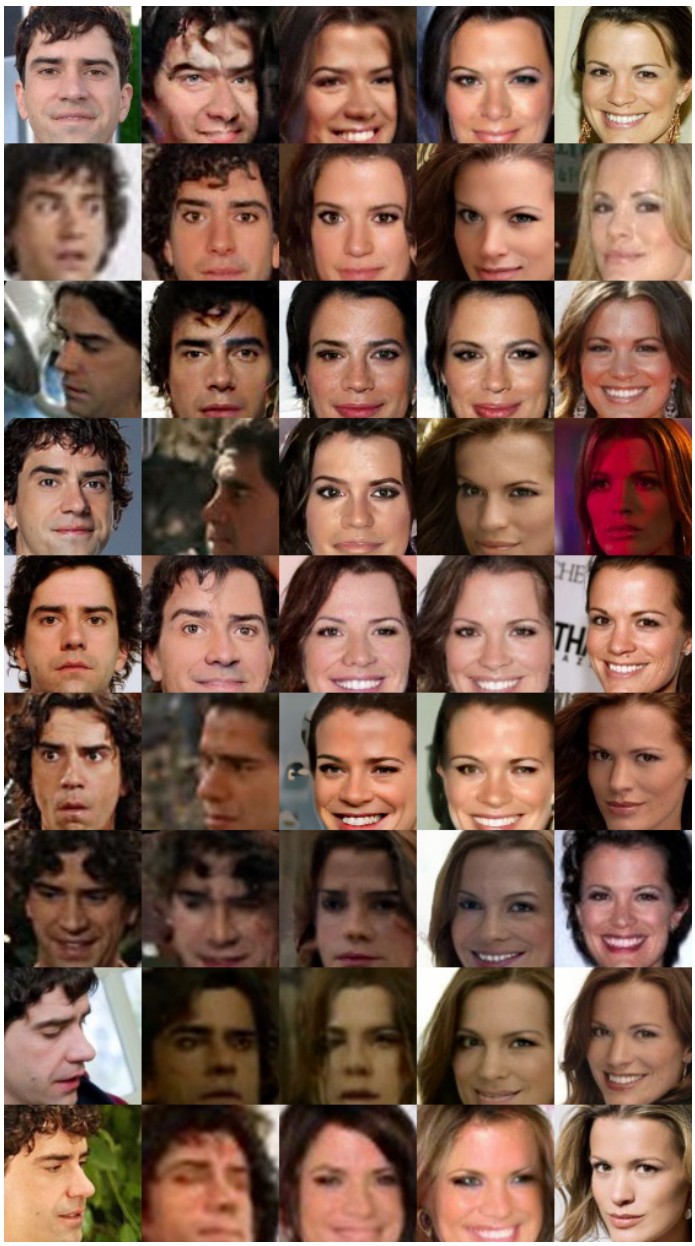

Figure 8: Small-sized generator trained till it sees 335M images in different noise levels (∼700 Epochs). From left to right, the first column is variations of a random ID, 1, in the, $D^{orig}$, the second column is the recreation of the same ID in the first column using the generator when we set the corresponding conditions to 1. The last two columns are the same but for different IDs and the middle column representing the $D^{aug}$ sample.

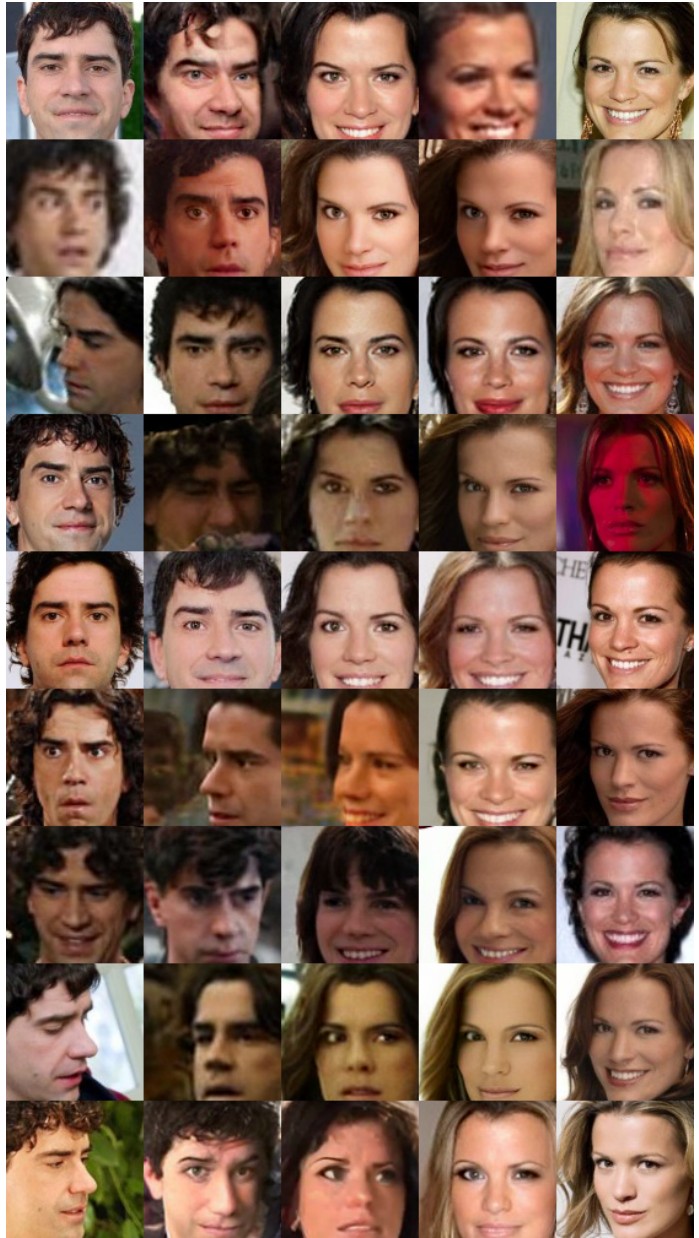

Figure 9: Medium-sized generator trained till it sees 335M images in different noise levels (∼700 Epochs). From left to right, the first column is variations of a random ID, 1, in the, $D^{orig}$, the second column is the recreation of the same ID in the first column using the generator when we set the corresponding conditions to 1. The last two columns are the same but for different IDs and the middle column representing the $D^{aug}$ sample.

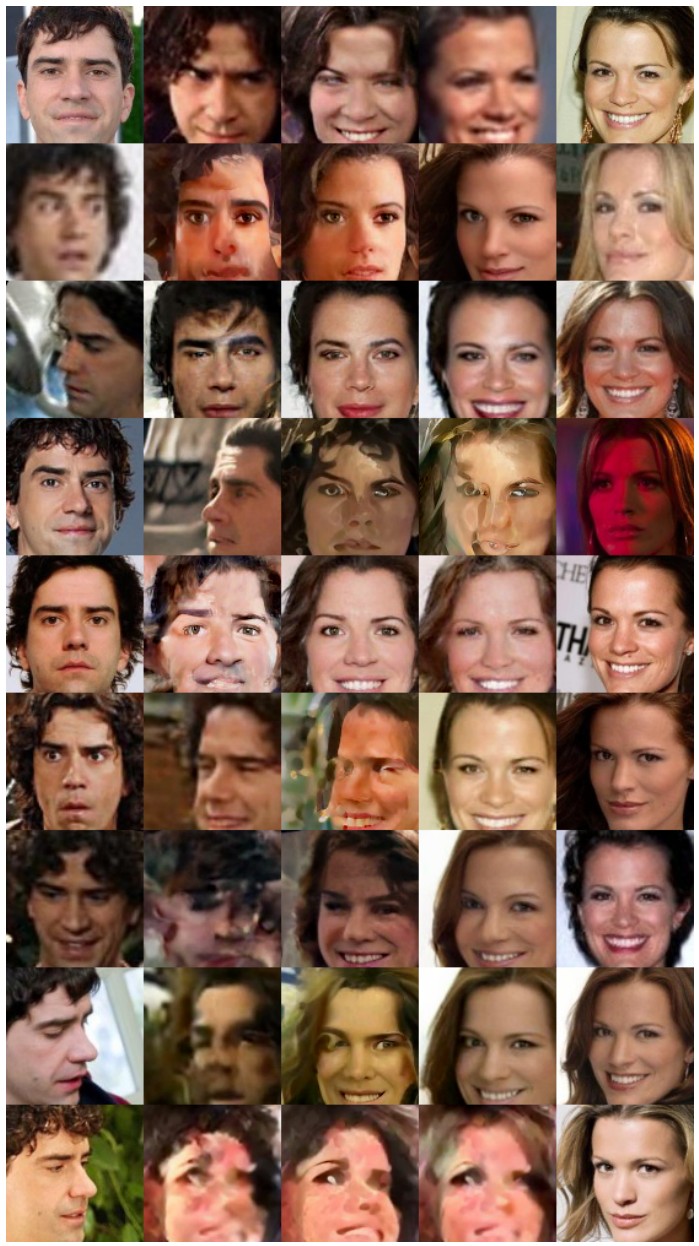

Figure 10: Medium-sized generator trained till it sees 805M images in different noise levels ($\sim$1500 Epochs). From left to right, the first column is variations of a random ID, 1, in the, $D^{orig}$, the second column is the recreation of the same ID in the first column using the generator when we set the corresponding conditions to 1. The last two columns are the same but for different IDs and the middle column representing the $D^{aug}$ sample.

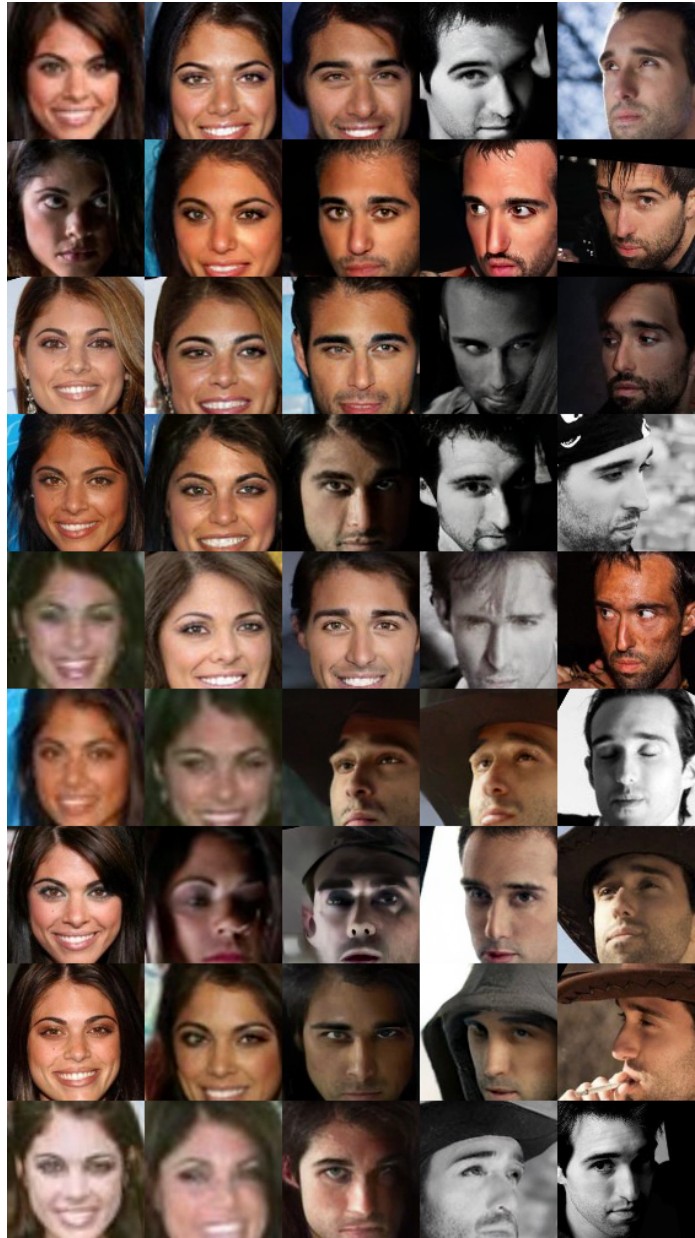

Figure 11: Medium-sized generator trained for till it sees 335M images in different noise levels (∼700 Epochs) for different IDs. From left to right, the first column is variations of a random ID, 1, in the, $D^{\text{orig}}$, the second column is the recreation of the same ID in the first column using the generator when we set the corresponding conditions to 1. The last two columns are the same but for different IDs and the middle column representing the $D^{\text{aug}}$ sample.

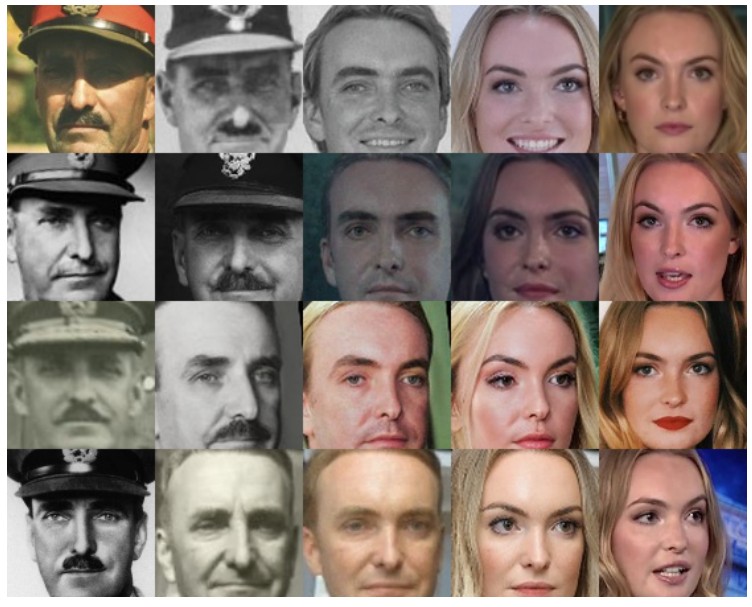

(a) IDs 115 and 2668

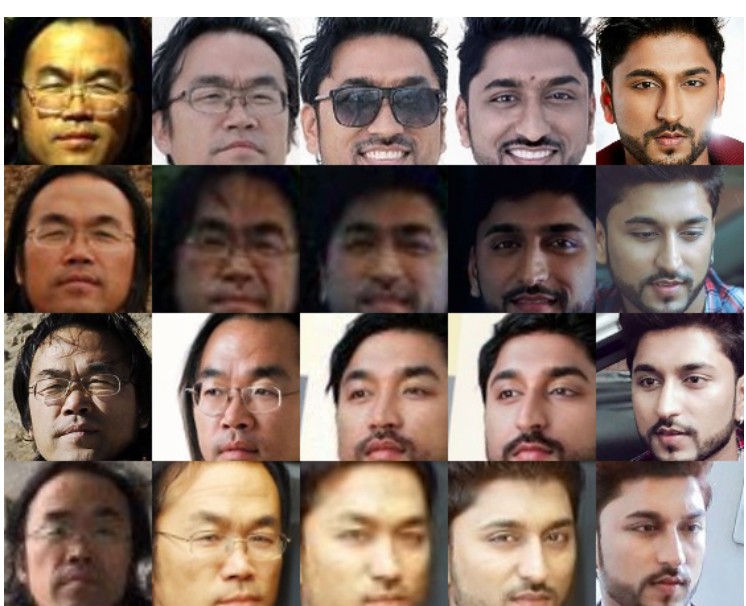

(b) IDs 760 and 1297

Figure 12: Samples from a small-sized pixel space EDM generator trained on WebFace160K for about 31M training steps ($\sim$200 Epochs). From left to right, the first column is variations of a random ID, 1, in the, $D^{\text{orig}}$, the second column is the recreation of the same ID in the first column using the generator when we set the corresponding conditions to 1. The last two columns are the same but for different IDs and the middle column represents the $D^{\text{aug}}$ sample.

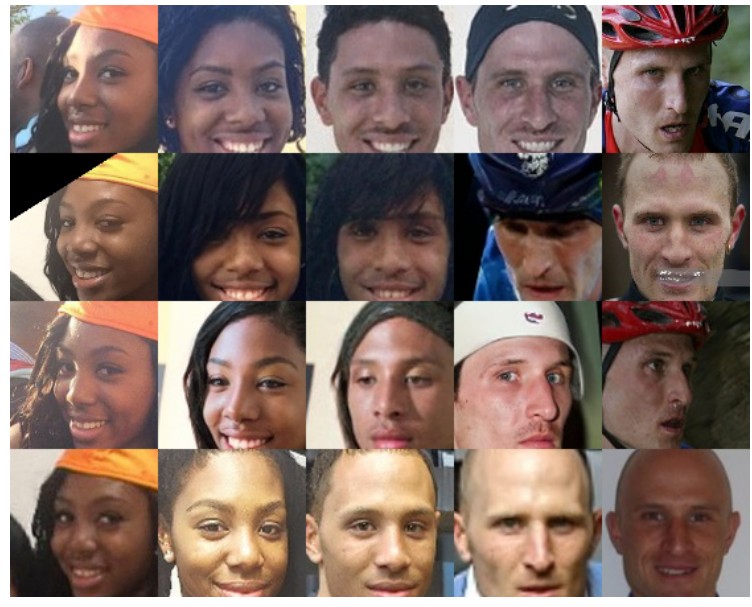

(a) IDs 2299 and 8574

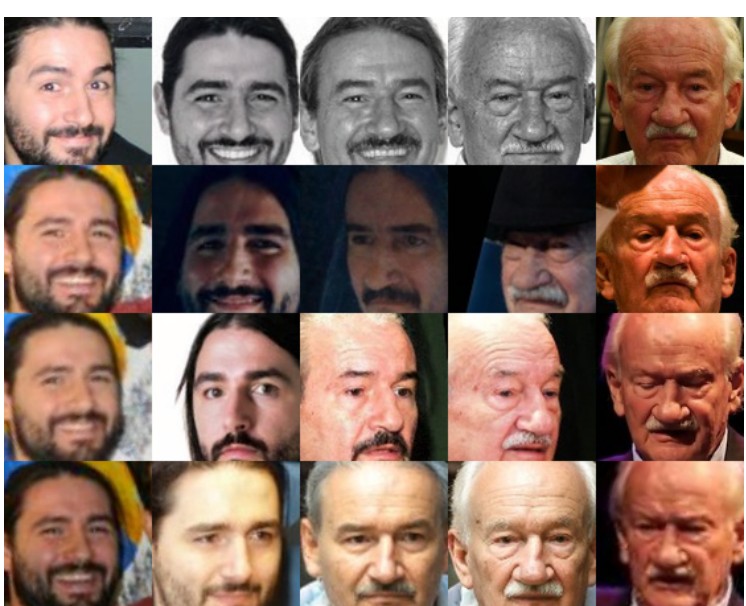

(b) IDs 7858 and 8434

Figure 13: Samples from a small-sized pixel space EDM generator trained on WebFace160K for about 31M training steps ($\sim$200 Epochs). From left to right, the first column is variations of a random ID, 1, in the, $D^{\mathrm{orig}}$, the second column is the recreation of the same ID in the first column using the generator when we set the corresponding conditions to 1. The last two columns are the same but for different IDs and the middle column representing the $D^{\mathrm{aug}}$ sample.

## Impact Statement

In our approach, we introduce a novel technique that leverages generative models to further improve state-of-the-art (SOTA) facial recognition (FR) systems, as demonstrated on publicly available medium-sized datasets. However, these same FR systems can inadvertently facilitate unauthorized identity preservation in deepfakes and other forms of fraudulent media when attackers mimic individuals without their consent.

While our primary objective is to address privacy concerns and informed consent in training FR systems, the resulting performance gains could also enhance deepfake quality.

