# OpenReview forum: "AugGen: Synthetic Augmentation using Diffusion Models Can Improve Recognition"
_NeurIPS.cc/2025/Conference — NeurIPS 2025 poster_

### Official Review · Reviewer_swWZ · 2025-06-30

**Clarity:** 3
**Significance:** 3
**Originality:** 3
**Rating:** 4
**Confidence:** 4

**Summary:**

This paper presents AugGen, a simple yet effective framework for data augmentation using synthetic images generated from a conditional diffusion model. The method aims to enhance discriminative models, particularly in face recognition, by introducing pseudo-novel classes constructed through weighted linear combinations of one-hot class condition vectors. The generated images are filtered using a feature-space-based criterion that maximizes inter-class dissimilarity and intra-class consistency. These selected synthetic samples are then mixed with real data to form an augmented training set. The approach is validated across a wide range of face recognition benchmarks. Extensive experiments show that AugGen achieves performance improvements comparable to architectural modifications or real-data expansion, highlighting its practical value in low-resource scenarios.

**Questions:**

Please refer to the Weaknesses section for additional context. Below are three questions and suggestions that target specific aspects of the current design and presentation, which, if addressed, could improve the clarity and impact of the work:

1. Can the heuristic search over (\alpha, \beta) be made learnable or end-to-end?
The proposed framework relies on manually enumerating mixing weights and selecting candidates based on a non-differentiable score. Have the authors considered making the weight selection process learnable, for example through a soft parameterization of the class mixing vector or meta-learning over mixing coefficients? A demonstration of a differentiable or learnable variant of the sampling mechanism could significantly enhance the method’s generality and scalability, and would strengthen the contribution.

2. Is diffusion necessary for the framework, or could other generators be used?
The current method is built on conditional diffusion models, but the core idea—class mixing and sample selection—appears generator-agnostic. Could the authors comment on whether similar results would hold with GANs, VAEs, or pretrained generative models like StyleGAN? An ablation or discussion on the necessity of diffusion models versus alternative generators would clarify the scope and portability of the approach.

3. Clarify presentation inconsistencies and formatting choices
Some parts of the paper, particularly in the methodology section, suffer from overly long paragraphs and occasional typographical issues (e.g., “synthetic images.ss”). Additionally, the use of boldface for variables like \mathbf{X} and \mathbf{Z} is inconsistent and may confuse readers, especially since they do not appear to carry a distinct semantic meaning. Clarifying these presentation details would improve the clarity of the paper.

**Ethical Concerns:**

["NO or VERY MINOR ethics concerns only"]

**Final Justification:**

I appreciate the authors’ detailed reply. They clarified why they stuck with grid search and why diffusion models fit best here. It’s also good that they’ll fix the formatting issues. But I still feel the manual search limits the method’s flexibility, and there’s not a big leap in technical novelty. Overall, it’s a solid paper with useful insights, but my original concerns mean I’m keeping my initial score.

**Limitations:**

Yes

**Quality:**

3

**Strengths And Weaknesses:**

Strengths:
1. Strong motivation with partially inconsistent presentation
The motivation behind this work is compelling and well-argued: the authors aim to enhance discriminative model performance by leveraging synthetic augmentation without relying on external models or auxiliary data. This is a relevant and timely objective, particularly in privacy-sensitive domains such as face recognition.

2. Simple yet practical augmentation strategy
The paper proposes a lightweight, reproducible strategy for synthetic data augmentation by generating pseudo-novel classes through linear interpolation of class condition vectors in a conditional diffusion model. This approach avoids the use of external models or labelers, and instead relies solely on the internal structure of the training data and the generative model itself. While the concept of interpolating between classes is not entirely new, the authors effectively adapt it into a generative augmentation pipeline that is particularly well-suited for scenarios where acquiring or annotating real data is costly or constrained by privacy concerns. The fact that the method can be implemented without complex architecture changes or training objectives makes it attractive for deployment in real-world face recognition systems and potentially other classification settings.

3. Strong empirical performance with extensive evaluation
The experimental results are a notable strength of the paper. The authors evaluate their method across a diverse set of face recognition benchmarks. These datasets span both high- and low-quality image domains, and the consistent improvements over baseline methods demonstrate the robustness of the proposed strategy.  Furthermore, the paper includes comprehensive ablation studies that analyze the effects of mixing ratios, pseudo-class selection, and augmentation volume.

Weaknesses:
1. Limited methodological novelty
While the overall system is well-designed and thoughtfully implemented, the core components—namely, class interpolation through condition vector mixing and heuristic sample selection based on feature-space metrics—are primarily composed of existing ideas. The paper focuses on a clean and pragmatic integration of known techniques, rather than introducing new loss functions, training paradigms, or theoretical formulations. As such, its contribution lies more in empirical innovation and practical value than in conceptual novelty.

2. Heuristic and non-differentiable design
The current pipeline depends on a manual, non-learnable search over class pairs and mixing coefficients (\alpha, \beta), followed by a scoring mechanism based on hand-crafted similarity and dissimilarity metrics. This makes the system less flexible for integration into end-to-end differentiable learning frameworks. Moreover, such a heuristic process may become inefficient or brittle when scaling to larger class sets or tasks requiring dynamic sample generation.

3. Minor presentation issues
Although the paper is generally clear in its motivation and organization, there are several presentational inconsistencies that hinder readability. Some paragraphs—especially in the methodology section—are excessively long and would benefit from better structural segmentation. In addition, there are typographical issues (e.g., “with synthetic images.ss”) and formatting inconsistencies, such as the unexplained use of boldface for variables like \mathbf{X} and \mathbf{Z}, despite no distinction from their standard forms. While these issues do not detract from the technical content, addressing them would improve clarity and overall polish.

---

> ### Author Rebuttal · Authors · 2025-07-27
>
> We sincerley thank the reviewer for the detailed feedback and suggestions, and also for the time and effort invested in reviewing and improving our manuscript.
>
> -------------
> ## Weakness 1: Novelty ##
> We agree that a primary contribution of our work lies in the novel system design and its empirical validation, rather than the invention of entirely new standalone components. Yet we believe that this constitutes a significant addition to the literature. Our work demonstrates that a carefully designed and pragmatic integration of existing techniques can surprisingly outperform more complex approaches. As we show in our response to Q1 (below), our simple approach surpasses a much more intricate, end-to-end guided alternative. We argue that demonstrating that simplicity can be more effective is a valuable contribution, especially for practical applications in a field like face recognition where reliability and reproducibility are paramount.
>
>
> -------------
> ## Question 1/Weakness 2: Why grid search and heuristic? ##
> We initially attempted to optimize $\alpha$ and $\beta$ with respect to our metric $m_{total}$ (which is differentiable via $f_{\theta_{dis}}$), but the unstable gradients and the multi-step nature of diffusion inference made this approach diverge. We explored alternative techniques, including Classifier-Guidance [3R3] (using the discriminator to guide the generator) and GGDPM [3R2] but results were not promising.
>
> Therefore, we adopted a simple grid search, which is simple, stable, and generalizable across generator types (e.g., multi-step autoregressive models), where gradient-based guidance is often not applicable [3R6].
>
> To demonstrate this, we present some parts of our results from experiments with classifier guidance, where we encourage the denoising process to produce samples that lie between certain classes in the discriminator's embedding space. The update rule is:
> $$ \tilde S\_\theta = S\_\theta - \lambda\_t \mathbf{g}\_t, \qquad \mathbf{g}\_t = \nabla\_{\mathbf{x}\_t} \mathcal{L}(f\_{\theta\_{dis}}(\mathbf{x}\_t),\bar{\mathbf{e}}) $$
> where $\mathbf{g}\_t$ is the gradient guiding the sample $\mathbf{x}\_t$ towards a target embedding $\bar{\mathbf{e}}$ via $f\_{\theta\_{dis}}$. The rest of the notation follows Section 3.
> Despite some effort for hyperparameter tuning, we could not exceed the performance of our simpler `AugGen` approach.
>
> | Dataset | $n^r$ | $n^s$ | B-1e-5 | C-1e-6 | C-1e-5 | AgeDB | CALFW | TR5 |
> | :--- | :--- | :--- | :--- | :--- | :--- | :--- | :--- | :--- |
> | WebFace160K | 0.16M | – | 72.34 | 70.59 | 78.11 | 89.75 | 92.11 | 67.06 |
> | WebFace160K (AugGen) | 0.16M | 0.2M | **77.45** | **78.60** | **84.03** | **92.95** | **93.58** | **67.46** |
> | WebFace160K (AugGen **+ Classifier Guidance**) | 0.16M | 0.2M | 75.05 | 75.89 | 82.16 | 92.75 | 93.28 | 67.22 |
>
> As the table shows, even with classifier guidance added, the performance did not surpass the baseline `AugGen`.
>
> ### Optimizing via few-step distilled variants ###
> One natural difficulty for end-to-end optimization is the multi-step inference process. We tested recent methods that distill generation into a few steps (e.g., Adversarial Score Identity Distillation [3R4, 3R5]). While distillation improved the performance of the discriminator trained the on reproduced data ($D^{\mathrm{repro}}$), applying guidance or `AugGen` to the few-step distilled models resulted in a significant drop in image quality, likely due to mode collapse from the adversarial distillation loss (Please see the answer to the next question too.)
>
> **We propose to include these negative results and failure cases in our Appendix to guide future work.**
>
>
> -------------
> ## Question 2: Can we use different generator types? ##
>
> ### VAEs, Diffusion and Flows ###
> This is a very good question. Theoretically, both VAEs and Diffusion Models train a generator with a maximum likelihood or ELBO objective; for a detailed derivation, please see [3R1]. We chose to use a diffusion model primarily because the methodology is more mature, and there are stable empirical procedures for both training (e.g., SNR-based weighting for high-resolution images) and inference (e.g., faster samplers like DPM-v3).
>
> The same can be said for Flow Matching. More specifically, methods like Gaussian Flow Matching (used in Flux and SD3) can be directly formulated as a diffusion model under a v-prediction parameterization.
>
> ### GANs ###
> The main difference lies with GANs, whose objective is not formulated as an ELBO or ML. During our experimentation, we attempted to train StyleGANv3 from scratch on our datasets (CASIA-WebFace and WebFace160K), as no publicly available models were trained on these specific FR datasets, and we aimed to avoid any information leakage from external data like FFHQ. However, as it is well known, GANs are very difficult to train, and our training runs were divergent despite using the settings provided by the original authors. Furthermore, a primary concern with GANs is mode collapse. This makes them an unfavorable choice for our goal, which is to **explore out-of-distribution generation**. This is especially important for long-tailed datasets like CASIA-WebFace, where modes in the tail would likely not be recovered by a GAN-based generator.
>
> **We propose to include a discussion of these design choices in the Appendix**
>
> -------------
> ## Question 3 / Weakness 3: Simplification of text ##
> Thanks for spotting the typo [L126] and for the feedback. We reworked the presentation, especially Section 3, to improve readability.
>
> We will clarify the notation in the paper to distinguish vectors, matrices, and higher-order tensors, consistent with the implementation.
>
> -------------
> [3R1] Kingma, Diederik, and Ruiqi Gao. "Understanding diffusion objectives as the elbo with simple data augmentation." Advances in Neural Information Processing Systems 36 (2023): 65484-65516.
>
> [3R2] Guo, Yingqing, et al. "Gradient guidance for diffusion models: An optimization perspective." Advances in Neural Information Processing Systems 37 (2024): 90736-90770.
>
> [3R3] Dhariwal, Prafulla, and Alexander Nichol. "Diffusion models beat gans on image synthesis." Advances in neural information processing systems 34 (2021): 8780-8794.
>
> [3R4] Zhou, Mingyuan, et al. "Score identity distillation: Exponentially fast distillation of pretrained diffusion models for one-step generation." Forty-first International Conference on Machine Learning. 2024.
>
> [3R5] Zhou, Mingyuan, et al. "Adversarial score identity distillation: Rapidly surpassing the teacher in one step." arXiv preprint arXiv:2410.14919 (2024).
>
> [3R6] Yu, Qihang, et al. "Randomized autoregressive visual generation." arXiv preprint arXiv:2411.00776 (2024)

---

> > ### Comment · Reviewer_swWZ · 2025-08-05
> >
> > Thanks for the rebuttal. It’s a solid paper with useful insights. Please consider including the discussion in the final version.

---

> > > ### Author Response · Authors · 2025-08-06
> > >
> > > We sincerely thank the reviewer for the valuable suggestions and for carefully checking our rebuttal. We will include the discussions in the paper as promised.

---

### Official Review · Reviewer_pBfk · 2025-07-03

**Clarity:** 4
**Significance:** 3
**Originality:** 3
**Rating:** 4
**Confidence:** 3

**Summary:**

This paper presents AugGen, a self-contained synthetic enhancement technique. By strategically sampling from a class-conditional generative model trained exclusively on the target face recognition dataset, it generates synthetic samples without external resources, effectively boosting the model's discriminative ability. Evaluations on 8 FR benchmarks (including IJB-C and IJB-B) show that this approach achieves a 1-12% performance improvement, outperforming models trained solely on real data and state-of-the-art synthetic data generation methods. It uses less real data, and its gains often exceed those from architectural modifications, demonstrating the value of synthetic enhancement in data-constrained scenarios.

**Questions:**

Please see the Weakness part.

**Ethical Concerns:**

["NO or VERY MINOR ethics concerns only"]

**Final Justification:**

The detailed rebuttal has addressed my concerns. I maintain my original score of "4: BA".

**Limitations:**

yes

**Paper Formatting Concerns:**

None.

**Quality:**

3

**Strengths And Weaknesses:**

**Strength:**
1. The paper proposes a self-contained synthetic augmentation technique called AugGen. The mixed-class samples generated by it can significantly enhance the performance of FR benchmarks, fully demonstrating its effectiveness.
2. The writing and structural organization of the paper are good, and the comparative experiments are comprehensive.

**Weakness:**
1. As shown in Figure 2, the paper involves multiple stages, including Generator training and data generation. The overhead brought by these processes should be considered in practical application scenarios. It is suggested to supplement information on the training time and resource consumption of different stages.
2. State-of-the-art methods like GPT-4O and Qwen-VLo are capable of generating highly realistic face data. Could these be used to enhance the performance of FR? It would be beneficial to include some comparative analysis on this.
3. Table 2 in the paper shows that more forged samples are needed to achieve results comparable to real samples, but there is a lack of in-depth analysis on the reasons. Is it due to the interference of mixed identities in synthetic data, or should some strategies be introduced in training to handle real and synthetic data differently?

---

> ### Author Rebuttal · Authors · 2025-07-27
>
> We sincerely thank the reviewer for their thoughtful feedback and valuable suggestions, which have greatly helped us improve the manuscript.
>
>
> -------------
> ## Weakness 1: Compute cost ##
> We propose to add a discussion to the paper about this important aspect. The proposed method adds a non-trivial one-time training cost, but this is amortized as it yields a model that is both more accurate and more efficient at inference.
>
> We present a cost breakdown below. Augmenting the data lets the smaller IR-50 backbone outperform the much larger IR-101 model (~1.9×FLOPs and ~1.7×parameters) trained on the original data (Table. 1.)
> Crucially, our final model retains the low inference cost of the IR50 backbone while outperforming the IR101 model, which is vital for real-world deployment where cumulative inference costs quickly surpass the one-time training expense [2R3].
>
> | | Train Generator | Train IR50 on $D^{\mathrm{orig}}$ | Train IR50 on $D^{\mathrm{orig}} + D^{\mathrm{aug}}$ (Ours) | Train IR101 on $D^{\mathrm{orig}}$ |
> | :--- | :--- | :--- | :--- | :--- |
> | **GPU type** | 1x H100 | 4x 3090Ti | 4x 3090Ti | 4x 3090Ti |
> | **Wall time (h)** | 42.2 | 2.54 | 4.1 | 5.6 |
> | **Average perf.** | N/A | 27.42 ± 0.92| **32.63 ± 2.20** | 27.24 ± 1.07 |
>
> *Variances are calculated as the pooled standard deviation from the results reported in Table 1.*
>
> This demonstrates a favorable trade-off: we accept a higher, fixed training cost to produce a superior model that is cheaper to deploy. We will release our source code and trained models to further mitigate this cost for the community.
>
>
> -------------
> ## Weakness 2: Usage of generalist models like GPT-4o, FLUX, Gemini for dataset generation ##
> This is an important aspect that we propose to clarify in the introduction section. State-of-the-art generalist models may generate higher-quality images, but they come with significant constraints (see below). This is the key motivation for our development of a self-contained method (independent of external datasets, commercial APIs, or existing models).
>
> - **License restrictions.** Generalist models like GPT-4o and Gemini have restrictive usage policies [2R1, 2R2] that prevent their use in sensitive or commercial applications like face recognition.
>
> - **Unknown training data & consent issues.** Many generalist models are trained on private data, where subject consent cannot be guaranteed. This poses a major concern for face recognition systems, medical applications, and other sensitive use cases—an issue our work explicitly avoids.
>
>
> -------------
> ## Weakness 3: Why is synthetic data less sample-efficient than real data? ##
> This is an excellent question. Our approach **does not** claim to introduce entirely **new information**; rather, it strategically re-uses information from the original dataset to find hard samples the discriminator may have missed during initial training. We offer two primary reasons why more synthetic samples are beneficial.
>
> 1.  **Recovering Missed Information.** The generator's role is to explore the data distribution and produce hard-to-classify samples. Presenting a larger volume of these generated samples naturally increases the probability of exposing the discriminator to these challenging examples, thereby improving its robustness.
>
> 2.  **Generator Imperfection.** As shown by the performance of a model trained only on reproduced data ($D^{\mathrm{repro}}$) versus the original data ($D^{\mathrm{orig}}$) (i.e., Table.1) our generator does not perfectly replicate the real data distribution. This slight imperfection means each synthetic sample may carry less informational value than a real one, necessitating a larger quantity of augmented data to inject the same amount of useful information.
>
> Ultimately, our results demonstrate that even state-of-the-art discriminator models (usually used in face recognition), which are already highly optimized, can be significantly improved without adding any new real datasets or pre-trained models.
>
> -------------
> [2R1] https://openai.com/policies/creating-images-and-videos-in-line-with-our-policies
>
> [2R2] https://ai.google.dev/gemini-api/terms#use-restrictions
>
> [2R3] https://www.breakingviews.com/columns/big-view/ai-boom-is-infrastructure-masquerading-software-2025-07-23/

---

> > ### Comment · Reviewer_pBfk · 2025-08-05
> >
> > Thank you to the author for the detailed rebuttal, which has resolved most of my concerns. Regarding the new synthetic methods like GPT-4o, the author mentioned application limitations. However, I am just curious about how different face generation methods and the realism of generated faces would impact the metrics in the paper.

---

> ### Author Response · Authors · 2025-08-06
> **Generalist models**
>
> We thank the reviewer for the thoughtful feedback and following up to our rebuttal. As requested, we present below additional experiments with state-of-the-art generative models.
>
> ## Open-weight model: Flux-1-Kontext-dev
> We tested ``Flux-1-Kontext-dev``, the best available open-weight image-to-image model currently available (see benchmarks at https://lmarena.ai/leaderboard/image-edit).
>
> Only a single input image is supported due to implementation constraints from the diffusers library (https://github.com/huggingface/diffusers/tree/main/src/diffusers/pipelines/flux). To preserve identity context, we concatenate between 1 and 6 images of two random identities (similar to AugGen) and test various prompts. One of the best-performing prompts was the following.
>
> > "*There are two columns in this image, each corresponds to images of a distinct identity. Generate a morphed image combining cues from both identities to create a new, perceptibly distinct identity.*"
>
> However, results were generally underwhelming. Even with our best prompts, the resulting images are mostly super-sampled versions of the input images, with very little variation in appearance or identity.
>
> ## OpenAI gpt-image-1
> We also evaluated OpenAI’s `gpt-image-1` (currently top performer on image-to-image tasks). We tested it on a small set of images, since cannot upload the whole dataset to a third-party server (we need to ask for permission from original data curators). We report metrics from a strong face recognition system (IR101 trained on WebFace4M):
>
> - **RID1 x RID2**: average unique similarities between images of two different original identities.
> - **RID1-RID2 intra**: average intra-class similarity within each original identity class.
> - **SID intra**: average intra-class similarity of generated synthetic images.
> - **SID x all RID1-RID2**: average similarity between synthetic and original images.
>
> | Metric                | Value            |
> |-----------------------|------------------|
> | RID1 x RID2           | 0.0350 ± 0.0800  |
> | RID1-RID2 intra       | 0.5334 ± 0.0042  |
> | SID intra             | 0.3938 ± 0.1568  |
> | SID x all RID1-RID2   | 0.1171 ± 0.0138  |
>
> Key observations:
> - `SID x all RID1-RID2` > `RID1 x RID2`. Synthetic samples resemble their sources, acting as hard examples.
> - `SID intra` > 0.3. Effective identity preservation in synthetic samples.
>
> These results suggest that `gpt-image-1` has some potential for generating synthetic images with similar effects as our approach. Its downside is the high cost (~USD 10 for 140 images in HQ) which severly limits its use at scale, compared to our approach. This is especially important since we showed in Table 2 (main paper) that a relatively large amount of synthetic data is needed to improve performance compared to real data.
>
>
> Please let us know if this answers your question.

---

> > ### Author Response · Authors · 2025-08-09
> >
> > Please let us know if you have any further questions. We appreciate your time and consideration.

---

### Official Review · Reviewer_Yi5o · 2025-07-03

**Clarity:** 3
**Significance:** 3
**Originality:** 3
**Rating:** 5
**Confidence:** 4

**Summary:**

This paper proposes a method that leverages an existing face recognition dataset to train a generative model, which is then used to synthesize images for training the face recognition model. The approach involves using the one-hot label of the recognition model as a conditional input to the generator, and interpolating between existing training data to create face images of intermediate identities as new data. These synthesized samples are then used to train the face recognition model, leading to improved recognition performance.

**Questions:**

1.	If the cosine similarity of two human faces is small but the difference is huge, is the interpolated face still representative?
2.	If we interpolate two faces that are very similar, will the resulting faces still be distinguishable?

**Ethical Concerns:**

["NO or VERY MINOR ethics concerns only"]

**Final Justification:**

The rebuttal has addressed most of my concerns. And after reading reviews of the other reviewers, I am willing to raise my rating.

**Limitations:**

yes

**Quality:**

3

**Strengths And Weaknesses:**

Strengths :
1.	The paper proposes synthesizing new identities by interpolating between two existing identities based on the available data. This approach enables the generation of novel identities, and an algorithm is introduced to evaluate whether the generated data is suitable for recognition training.
2.	The method utilizes one-hot labels as input to control image generation. This relies on the training of the original recognition model (M_ori) and uses the original data to train the generative model. Additionally, the one-hot representation allows for interpolation between two identities.
3.	The experimental results are promising and demonstrate the effectiveness of the proposed approach.

 Weaknesses:
1.	The paper uses one-hot vectors as conditional input, but it is unclear how interpolated faces are generated when values between two identities (i.e., numbers not seen during training) are used. The paper does not clearly explain whether there are additional constraints during training to address this issue.
2.	The experiments are conducted on a dataset with strong training data. It is unclear whether the proposed method would still be effective when the dataset contains only a single image per identity. The effectiveness of privacy protection in such cases is also not discussed. Alternatively, as a data augmentation method, experiments should be conducted on a larger dataset (such as MS1M).
3.	Although new identities are synthesized, these identities are interpolated from real identities. This approach does not demonstrate that truly new identities are generated, and thus it is unclear whether privacy is actually protected.
4.	When optimizing αα and ββ in the paper, it is only performed on a subset of the identities. The paper does not explain how these identities are selected. The search space would be larger if all identities were considered, and more identity data could potentially be synthesized.
5.	It is recommended to standardize the mathematical notation throughout the paper.

---

> ### Author Rebuttal · Authors · 2025-07-27
>
> We sincerely thank the reviewer for the thoughtful feedback and suggestions, and for the time invested in reviewing and improving our manuscript.
>
> We highlight that the paper contains **appendices** that provide additional support for our claims and address most of the reviewer's questions. We included these additional results in the final version.
>
> -------------
> ## Weakness 1 ##
>
> **One-hot conditoning.** We generate interpolated faces using soft conditioning vectors (i.e. interpolation between one-hot vectors). Formally, let $\mathbf{c}$ be the condition vector and $\mathbf{W}$ be the **learned** condition embedding matrix. The selection of an embedding is: $\mathbf{c}^T \mathbf{W}$. If we set $\mathbf{c}$ as a soft vector such as $[0.7, 0, 0.7]$, the resulting condition vector is a linear combination: $0.7 \times \mathbf{W}_1 + 0.7 \times \mathbf{W}_3$.
>
> **Constraint on condition space.** Despite being trained end-to-end using only one-hot condition vectors and without explicit regularization on condition space, the learned condition embeddings exhibit meaningful structure. Our method effectively exploits this structure to generalize to unseen conditioning vectors.
>
> **We will include this clarification in the final version.**
>
> -------------
> ## Weakness 2: Training dataset ##
> **Why CASIA-WebFace and WebFace160K?** We selected CASIA-WebFace to ensure a fair comparison with prior synthetic face generation methods (e.g., DCFace, VariFace, and Vec2Face), as these methods frequently use it as a training dataset. Our approach, which relies solely on CASIA-WebFace, notably surpasses these previous methods that typically incorporate additional datasets or sources of information. However, due to identified inconsistencies within CASIA-WebFace—detailed in our paper—we introduced WebFace160K, a carefully curated subset of WebFace4M. We employed this new dataset to further validate our method's effectiveness and recommend its adoption by the community.
>
> **Will it work with a single image per identity?** To effectively capture identity-preserving features, every face recognition training requires multiple images per identity. Our method trains both the generator and discriminator from scratch, making this requirement particularly crucial for learning meaningful identity representations—especially given the margin-based discriminator loss. Similarly, the generator’s condition space benefits from multiple samples per identity. We indirectly evaluated scenarios with few images per identity using CASIA-WebFace, which has a heavy-tailed distribution (Table 4), including identities with as few as two images. However, our approach may be less effective when strictly limited to a single image per identity, due to insufficient identity-specific variation. In fact, any face recognition training, especially those using margin-based losses, would suffer under such a strict constraint, as margin losses fundamentally rely on multiple examples per identity to enforce their key design principles.
>
> **Privacy protection.** We claim to mitigate privacy concerns primarily because, using a SOTA discriminator and our method, we can reduce the dataset size required to train a SOTA discriminator (with same performance) by up to $40$%  (Table 2), thereby mitigating privacy risks associated with collecting large datasets.
>
>
> **Why not MS1M or larger datasets?**
> Please note that large datasets like MS1M have been **retracted** due to privacy concerns, as they were often collected without consent from the identities [1R1]. Thus, for legal and reproducibility reasons, we refrained from using this dataset, despite its continued online availability. Additionally, while we validated our method on another dataset-a subset of WebFace4M- (for first time in Synthetic Dataset Generation community), training a generator directly on the full-scale dataset was beyond our available computational resources.
>
> -------------
> ## Weakness 3: Novel identities or hard samples? ##
>
> This is an excellent question. Our goal is to generate synthetic samples that are maximally beneficial for training the discriminator. While the synthesized identities have some similarities to their sources, we have demonstrated qualitatively that they represent distinct identities. By design, the generated samples are similar to their source classes. This similarity makes them effective **hard samples** for the discriminator. The discriminator must not only distinguish between different source classes but also differentiate between real images and our generated samples, which share many cues with their sources yet are distinct.
>
> Our results show that, with a limited amount of data, our approach achieves a performance boost equivalent to adding **1.7x** the original dataset's volume (Table 2). This means one would need to add 70% more real data to observe the same performance gain achieved by our method. thereby, as mentioned previously, mitigating privacy risks associated with collecting large datasets.
>
> -------------
> ## Weakness 4: Selection process for finding $\alpha$ and $\beta$ ##
> To determine the optimal $\alpha$ and $\beta$, we randomly selected 1,000 pairs of identities from the $\binom{N}{2}$ possible combinations [L246], where $N$ is the number of classes in the source dataset (e.g., CASIA-WebFace or our WebFace160K). We tested different random sets of pairs and observed a negligible impact on the optimal values, which remained $(\alpha=0.7, \beta=0.7)$ with a step resolution of 0.1. We added a short ablation study for this process in the final version.
>
> -------------
> ## Weakness 5: Notations ##
> Thanks! We fixed the notations to be coherent throughout the paper.
>
>
> -------------
> ## Question 1: Representative face images in most scenarios? ##
> Qualitatively and quantitatively (please refer to the next question), we observe that the generated images are plausible both when the source identities are similar (high cosine similarity) and when they are very different, such as having distinct ethnicities or ages (low cosine similarity). We provide examples for both cases in Figures 12 and 13 of our appendix. For instance, in Figure 13a, the source identities differ significantly in gender, ethnicity, and age. Nevertheless, the AugGen samples (third column) are not only plausible faces but also maintain strong identity consistency down the column, indicating a stable, novel identity has been generated.
>
> -------------
> ## Question 2: Similar and different face interpolation ##
> This is a very interesting question. Through extensive exploration of identity selection strategies, we found that mixing identities that are **further apart** in the discriminator's embedding space yields a greater performance improvement. When we perform mixing (using `Alg. 2`) on identities that are already similar, the performance gain is lower compared to when we select identities that are highly distinguishable from each other. We have summarized these results in the following table:
>
> | Dataset | Source ID Selection Strategy | $n^r$ | $n^s$| B-1e-5 | C-1e-6 | C-1e-5 | AgeDB | CALFW | TR5 |
> | :--- | :--- | :--- | :--- | :--- | :--- | :--- | :--- | :--- | :--- |
> | WebFace160K | Baseline (No Augmentation) | 0.16M | - | 72.34 | 70.59 | 78.11 | 89.75 | 92.11 | 67.06 |
> | WebFace160K | Mix **Similar** IDs (Closer in Embedding Space) | 0.16M | 0.2M | 73.11 | 71.86 | 81.00 | 91.85 | 92.81 | 66.82 |
> | WebFace160K | Mix **Dissimilar** IDs (Distant in Embedding Space) | 0.16M | 0.2M | **77.45** | **78.60** | **84.03** | **92.95** | **93.58** | **67.46** |
>
>
> **We propose to include a more detailed analysis of this finding in the appendix.**
>
>
> -------------
> [1R1] https://www.vice.com/en/article/microsoft-deleted-a-facial-recognition-database-but-its-not-dead/

---

> ### Author Response · Authors · 2025-08-07
>
> Please let us know if you have any additional questions. Thank you for your time and consideration.

---

### Note · Authors · 2025-08-14

We thank reviewers for their constructive feedback, which has been very helpful to clarify and strengthen the paper.
## Key highlights & contributions
Our work introduces AugGen, a novel technique with several key contributions:
*   **Significant performance improvement.** Consistent **1-12% performance gain across 8 FR benchmarks**, equivalent to adding **1.7x more real data**, thus showing significant value in data-limited scenarios.
*   **Practical, self-contained solution.** Works without external datasets or pre-trained models, directly addressing privacy and resource constraints inherent to applications like face recognition.
*   **Principled, privacy-first paradigm.** Proposes and validates a self-contained paradigm prioritizing privacy. To our knowledge, we are the first to show at scale that synergy of generator and discriminator can outperform more complex, data-hungry methods.
*   **New curated benchmark.** Introduce **WebFace160K** to resolve the demonstrated inconsistencies of the de facto standard, CASIA-WebFace, and a new benchmarking paradigm.
*   **Critique of evaluation metrics.** We show that **common generative metrics (FD,KID, ...) correlate poorly with downstream performance**, highlighting the need for better evaluation proxies for synthetic data.

## Rebuttal & enhancements
We believe we have addressed all the reviewers' main points, resulting in a stronger paper.

*   **Context & efficiency (pbfk).** Added experiments with SOTA models `gpt-image-1`, `Flux` and a **cost/time analysis**, confirming our approach is more practical and efficient. Our smaller IR-50 model outperforms the larger IR-101 (~1.9x FLOPs).
*   **Heuristic validation (swwz, yi5o).** Validated our simple heuristic by showing a more complex guided approach underperforms. We also added a discussion on design choices (e.g., Diffusion vs. StyleGANs) to clarify our methodology.
*   **Core technical claims (Yi5o).** Clarified technical details and added an ablation study proving that mixing *more dissimilar* identities yields greater performance gains (**77.45%** vs. **73.11%** TAR), offering additional gain/insights.
*   **Presentation (swwz, yi5o).** We improved the notations and structure according to the reviewers' suggestions.

We again thank the AC and reviewers for their time and consideration.

---

### Decision · Program_Chairs · 2025-09-17

**Decision:**

Accept (poster)

**Comment:**

This paper addresses privacy and ethical challenges through synthetic data generation. The method creates new identities by interpolating between existing ones and uses one-hot labels to guide image generation. Experiments demonstrate that it outperforms models trained only on real data and other state-of-the-art synthetic data methods.

All reviewers recommend acceptance, and the paper is expected to have broad community interest. Therefore, it is recommended for acceptance.